# ALIGNDIFF: ALIGNING DIFFUSION MODELS FOR GENERAL FEW-SHOT SEGMENTATION

## ABSTRACT

Text-to-image diffusion models have shown remarkable success in synthesizing photo-realistic images. Apart from creative applications, can we use such models to synthesize samples that aid the few-shot training of discriminative models? In this work, we propose AlignDiff, a general framework for synthesizing training images and masks for few-shot segmentation. We identify *three levels of misalignments* that arise when utilizing pre-trained diffusion models in segmentation tasks. These misalignments need to be addressed to create realistic training samples and align the synthetic data distribution with the real training distribution: 1) instance-level misalignment, where generated samples fail to be consistent with the target task (*e.g.,* specific texture or out-of-distribution generation of rare categories); 2) scene-level misalignment, where synthetic samples are often object-centric and fail to represent realistic scene layouts with multiple objects; and 3) annotation-level misalignment, where diffusion models are limited to generating images without pixel-level annotations. AlignDiff overcomes these challenges by leveraging a few real samples to *guide* the generation, thus improving novel IoU over baseline methods in generalized few-shot semantic segmentation on Pascal-5$^i$, COCO-20$^i$ by up to 80%. In addition, AlignDiff is capable of augmenting the learning of out-of-distribution categories on FSS-1000, while naïve diffusion model generates samples that diminish segmentation performance. The code will be released.

## 1 INTRODUCTION

Few-shot semantic segmentation has recently attracted increasing attention (Min et al., 2021; Tian et al., 2022; Qiu et al., 2023), given that it copes with the scarcity of (pixel-level) densely annotated data in practical scenarios. Existing efforts have primarily focused on either designing specialized architectures in low-data regimes (Tian et al., 2020; Fan et al., 2022) or employing data augmentation to produce slight variations of few provided data (Qiu et al., 2023). However, these methods struggle to improve the performance, as they ultimately rely on the few training samples that often do not faithfully represent the real data distribution. Such an issue is depicted in Fig. 1 with the comparison between the few-shot real training sample distribution and the real testing distribution. This paper overcomes this limitation by leveraging general-purpose text-to-image diffusion models to synthesize additional examples that are *not biased* to the few-shot real sample distribution, thereby aiding the training of few-shot segmentation models.

Trained on easy-to-obtain image-text pairs, large-scale text-to-image diffusion image generation models (Rombach et al., 2022; Saharia et al., 2022) have shown remarkable success in creative applications. Unfortunately, naïvely using diffusion models fails to generate examples that effectively benefit the training of segmentation models, due to the presence of **three-level misalignments** between the generated sample distribution and the underlying data distribution of the target task (Fig. 1). 1) **Instance-level:** simple text conditioning may not synthesize desired instances that are consistent with the target task scenario (*e.g.*, specific appearance). We define this issue as *Out-Of-Distribution* (OOD) generation, which means the distribution of generated samples is misaligned with the real data distribution. OOD generation tens to happen on rare categories (*e.g.*, Merlion in Fig. 1), leading to degenerate samples that diminish final segmentation performance. 2) **Scene-level:** while diffusion models typically synthesize object-centric images, the training samples used for segmentation tasks require composition of realistic *scene layout*. 3) **Annotation-level:** training

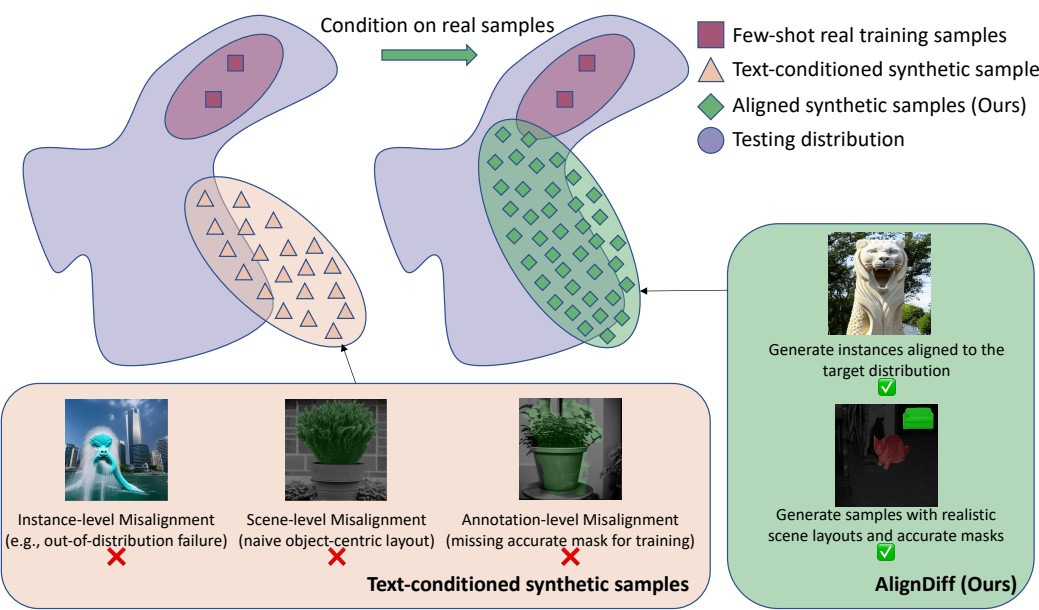

Figure 1: Illustration of issues in training segmentation models using a few real samples or pure text-conditioned synthetic/generated samples. The few provided real samples only represent a small subset of the target data distribution. On the other hand, text-conditioned synthetic samples from the synthetic distribution are misaligned to the target data distribution due to three types of misalignment. For instance-level misalignment, Stable Diffusion (Rombach et al., 2022) fails completely when tasked to generate images for rare categories such as the Merlion statue. Scene-level misalignment refers to the oversimplified object-centric scene layouts of synthetic examples, and annotation-level misalignment is caused by the limitations of diffusion models to generate only images. AlignDiff aims to address these issues by conditioning the generative process using a few real training samples, leading to an aligned synthetic distribution for training.

segmentation models needs accurate generation of *pixel-level mask annotation*, whereas existing diffusion models (Rombach et al., 2022) are limited to only generating images.

To address these challenges, we notice that the availability of a few real samples is a reasonable assumption to hold in practice. Thus, we introduce an **Align**ment **Diff**usion (AlignDiff) framework, with the key insight that *leverages the few real training data to guide the generation process* and aligns the generated distribution with the real distribution. Specifically, to address the instance-level misalignment, we propose *normalized masked textual inversion*. Our method learns an instance-specific word embedding, which ensures consistency with the given real examples. Empirically, our method works on OOD generation scenarios in which existing methods (Gal et al., 2022; Rombach et al., 2022) fail. Second, we propose to exploit copy-paste (Ghiasi et al., 2021) to combine synthesized instances with real images to generate training samples with more realistic layouts. Finally, to generate accurate pixel-level annotations, we design an efficient mask generation approach by gracefully relating to semi-supervised learning to condition the generative process on provided novel samples. We formulate the mask generation process as a process to refine noisy masks from a few examples of high-quality masks. Empirically, our approach is much more efficient than previous methods to extract masks from diffusion models (Wu et al., 2023) with on-par accuracy.

In sum, our **contributions** are as follows. **1)** To improve instance-level alignment and handle OOD generation, we propose normalized masked textual inversion that conditions the generation process using a few novel samples. On difficult out-of-distribution categories in the FSS-1000 dataset (Li et al., 2020), we show that samples synthesized with plain text conditioning diminish segmentation performance by over 10%. In contrast, AlignDiff increases the novel IoU by an additional 2 absolute points. **2)** To improve scene-level alignment, we propose to use real base samples via copy-paste to compose training samples with complex scene layouts. Compared to training with simple object-centric synthetic/generated samples, our composed samples drastically improve novel IoU by up to

80% for generalized few-shot segmentation on Pascal-5[i]. **3)** To improve annotation-level alignment, we take insights from semi-supervised learning (Wei et al., 2022) and propose a novel mask-generation pipeline to *guide* mask generation using a few novel samples. Compared to previous methods (Wu et al., 2023), our proposed method reduces the mask generation time from an average of 1 day per 2,000 images (Wu et al., 2023) to 1,000 seconds with similar mask quality.

## 2 RELATED WORK

**Semantic Segmentation.** Semantic segmentation is a dense vision task assigning a semantic label to each pixel in an image. Learning-based semantic segmentation methods can roughly be categorized into two paradigms: per-pixel classification and mask classification. Long et al. (2015) proposed the diagram for treating semantic segmentation as a per-pixel classification problem for Convolutional Neural Networks (CNNs). Later works (Zhao et al., 2017; Chen et al., 2017; 2018) then investigated different architectural improvements to further improve performance. More recently, an alternative paradigm, mask classification, was proposed by Cheng et al. (2021). This line of methods uses a detect-first-recognize-later paradigm. Since AlignDiff is a model-agnostic data synthesis method, works in designing network architectures for semantic segmentation are orthogonal to our method.

**Few-Shot Semantic Segmentation.** To allow segmentation models to operate in the low-data regime, Few-shot Semantic Segmentation (FSS) methods study how to predict segmentation masks of novel classes using only a few training examples of the novel class. Many methods (Wang et al., 2019; Tian et al., 2020; Fan et al., 2022; Min et al., 2021; Xu et al., 2023) and even specialized datasets such as FSS-1000 (Li et al., 2020) have been proposed to investigate this problem. Besides metric learning (Wang et al., 2019; Tian et al., 2020; Fan et al., 2022), recent works also exploit test-time optimization (Fan et al., 2022; Min et al., 2021) for few-shot segmentation.

Similar to conventional semantic segmentation, AlignDiff is orthogonal to work in FSS as it can be used to augment any proposed architecture in FSS. In this paper, we demonstrate the efficacy of AlignDiff on the FSS-1000 (Li et al., 2020) dataset. While plain diffusion models fail drastically on rare categories among the 1,000 categories in FSS-1000, we show that AlignDiff synthesizes samples that successfully improve the final performance of HSNet (Min et al., 2021).

**Generalized Few-Shot Semantic Segmentation.** Recently, generalized few-shot semantic segmentation (GFSS) had been proposed (Tian et al., 2022) as a more challenging task setting than vanilla few-shot segmentation methods. Compared to few-shot segmentation which produces novel-class-only binary masks, generalized few-shot segmentation tasks models to segment both base and novel classes within query images. Among recent works (Tian et al., 2022; Myers-Dean et al., 2021; Cermelli et al., 2021; Qiu et al., 2023), Tian et al. (2022) proposed to approach this problem using the test-time optimization scheme from few-shot segmentation, while later works (Myers-Dean et al., 2021; Cermelli et al., 2021; Qiu et al., 2023) found that fine-tuning the models on few provided data with continual learning techniques attain promising performance.

We apply AlignDiff to this challenging task setting by combining it with GAPS (Qiu et al., 2023), a recent work on GFSS. Empirically, AlignDiff consistently improves the segmentation performance on both the base and the novel classes on the widely adopted Pascal-5[i] and the COCO-20[i] dataset.

**Text-to-Image-Mask Generation.** There are some concurrent works that attempt to modify text-to-image synthesis models into text-to-image-*mask* synthesis models for training segmentation models. Li et al. (2023) proposed grounded diffusion (GD), a zero-shot segmenter for stable diffusion model (Rombach et al., 2022). However, the mask quality from GD (Li et al., 2023) is not ideal, as we quantitatively verified in experiments. Wu et al. (2023) proposed to use the intermediate attention maps in the diffusion models to generate coarse masks, which are then refined with a noise learning process. Though DiffuMask (Wu et al., 2023) generates masks of good quality, it requires heavy manual prompt engineering and its noise learning process is very time-consuming. Specifically, since DiffuMask requires full training of a segmentation model for cross-validation on every category, generating 2,000 image-mask pairs for a single category takes over 1 GPU day. In stark contrast, our AlignDiff generates high-quality masks with on-par accuracy as DiffuMask and much better efficiency - averaging 0.5 seconds per image. In addition, both GD (Li et al., 2023) and DiffuMask (Wu et al., 2023) suffer from the drawbacks in Fig. 1 and fail drastically on out-of-distribution categories.

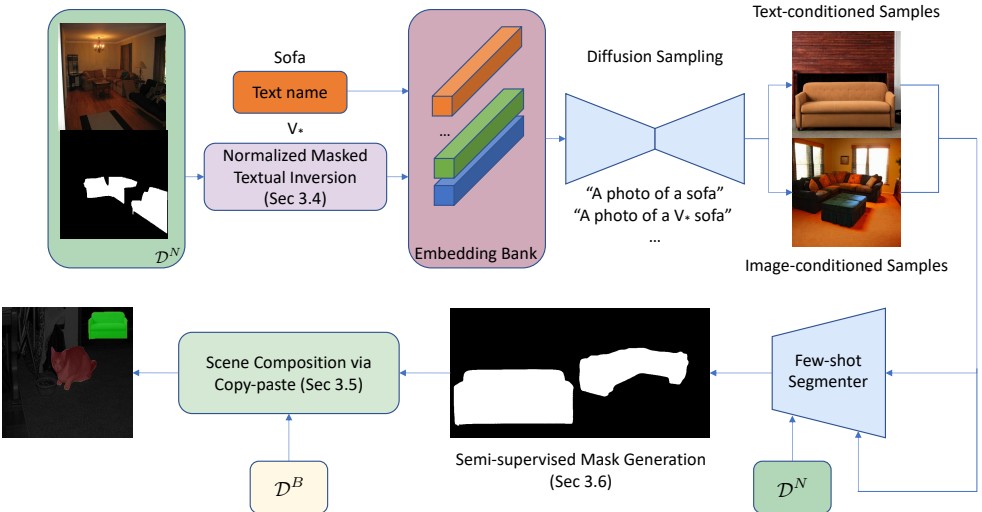

Figure 2: System overview of AlignDiff. We propose a Normalized Masked Textual Inversion method (Sec. 3.4) to condition the generative process based on as few as a single image, which inverses an image-mask pair to an instance-specific textual embedding. For pixel-level annotation generation, we propose a semi-supervised process that uses both synthetic samples and real samples to generate high-quality masks (Sec. 3.6). Finally, we compose training samples with realistic scene layouts by combining images from the base dataset $\mathcal{D}^B$ and synthetic samples (Sec. 3.5).

## 3 METHOD

### 3.1 PRELIMINARY: TEXT-TO-IMAGE DIFFUSION MODELS

Our method is based on Stable Diffusion (Rombach et al., 2022), which is an instance of the latent diffusion models that performs diffusion in the latent space with text conditioning. Given a text description $\mathbf{v} = (v_1, v_2, \ldots, v_n)$, where $v_i$ are token embedding encoded by a text encoding network from texts, starting from $X_T \sim \mathcal{N}(\mathbf{0}, \mathbf{I})$, the diffusion sampling process at each step is given by,

$$\mathbf{x}_{t-1} = \frac{1}{\sqrt{\alpha_t}} \left( \mathbf{x}_t - \frac{1 - \alpha_t}{\sqrt{1 - \bar{\alpha}_t}} \epsilon_\theta(\mathbf{x}_t, t, \mathbf{v}) \right) + \sigma_t \mathbf{z}, \tag{1}$$

where $\alpha_t$ and $\bar{\alpha}_t$ are constants regulating the denoising schedule, $\epsilon_\theta$ is a denoising U-Net (Ho et al., 2020) parameterized by $\theta$, $\sigma_t$ is a constant standard deviation, and $\mathbf{z} \sim \mathcal{N}(\mathbf{0}, \mathbf{I})$.

Let $\epsilon \sim \mathcal{N}(\mathbf{0}, \mathbf{I})$ and $x_t = \sqrt{\bar{\alpha}_t} x_0 + \sqrt{1 - \bar{\alpha}_t} \epsilon$ be a noisy version of original image $x_0$. The training process is then given by,

$$\mathbb{E}_{x_0, \epsilon \sim \mathcal{N}(\mathbf{0}, \mathbf{I}), t, \mathbf{v}} \left[ \left\| \epsilon - \epsilon_\theta(x_t, t, \mathbf{v}) \right\|_2^2 \right]. \tag{2}$$

### 3.2 PROBLEM FORMULATION

Let $\mathcal{X} \subset \mathbb{R}^{H \times W \times 3}$ be the set of RGB images, $\mathcal{C} \subset \mathbb{N}$ be a set of indexed categories, $\mathcal{M}_y$ be the name of the category, and $\mathcal{Y}^{\mathcal{C}} \subset \mathbb{R}^{H \times W \times |\mathcal{C}|}$ be a set of label masks. Following existing work on general few-shot semantic segmentation (Tian et al., 2022; Qiu et al., 2023; Min et al., 2021), we split the set of possible categories into $\mathcal{C}^B$, a set of base categories, and $\mathcal{C}^N$, a set of novel categories. We further assume that the base image-mask dataset $\mathcal{D}^B$ has abundant examples and the novel image-mask dataset $\mathcal{D}^N$ has only a few examples per category (*i.e.*, $|\mathcal{D}^B| \gg |\mathcal{D}^N|$).

The goal of this paper is to synthesize training examples of novel categories to augment few-shot learning. More formally, we want to design a synthesis function $\Phi(\mathcal{M}_y, \mathcal{D}^N, \mathcal{D}^B) \rightarrow \mathcal{X} \times \mathcal{Y}^{\mathcal{C}}$ that generates synthetic samples to augment $\mathcal{D}^N$ using $\mathcal{M}_y, \mathcal{D}^N$, and $\mathcal{D}^B$. Different from previous works to synthesize segmentation training samples (Wu et al., 2023; Li et al., 2023), which focuses on

using $\mathcal{M}_y$, our work focuses on *conditioning* the generative process using $\mathcal{D}^N$ and $\mathcal{D}^B$ to *align* the synthetic data distribution to the real data distribution (as illustrated in the green arrow in Fig. 1).

### 3.3 METHOD OVERVIEW

The insights and contributions of AlignDiff are centered at **how to condition the generative process using available real samples**. AlignDiff makes *full utilization* of abundant base training samples and a few novel samples, which leads to a distribution aligned with the real data distribution. Fig. 2 provides a high-level overview of AlignDiff. In Sec. 3.4, we discuss how to handle out-of-distribution generation for rare categories using a few novel samples via normalized masked textual inversion. In Sec. 3.5, we demonstrate how to create training samples with realistic scene layouts by fusing $\mathcal{D}^B$ with synthetic samples from $\Phi$. Finally, in Sec. 3.6, we describe how we relate mask generation to semi-supervised learning and describe a novel technique to use $\mathcal{D}^N$ to bootstrap such a process, which is shown to be much more efficient than existing method (Wu et al., 2023) with on-par accuracy.

### 3.4 OUT-OF-DISTRIBUTION GENERATION VIA NORMALIZED MASKED TEXTUAL INVERSION

Vanilla large-scale text-to-image diffusion models are not appropriate to be directly used to generate training samples. The reasons are two-fold: 1) text-to-image synthesis models may completely fail for OOD generation (*e.g.,* the Merlion statue in Fig. 1), and 2) text-conditioned synthetic samples may not be diverse. For instance, images generated with text prompts 'a photo of a sofa' share a similar pattern of straight upfront views of sofas (upper right corner of Fig. 2), which fails to capture the viewport variations and occlusion like the real-world samples (upper left corner of Fig. 2).

Existing methods (Wu et al., 2023) approach these two issues via prompt engineering, where hundreds of intra-class vocabularies are manually added to increase data diversity. However, such a method does not scale and it does not handle OOD generation. To address these challenges without handcrafted engineering, we propose *normalized masked textual inversion*. Given an image-mask pair $(x, y)$ and its class name $\mathcal{M}_y$, AlignDiff optimizes for instance-specific textual embeddings for provided instances, which serve as an implicit language description of properties of the novel object in $x$ (*e.g.,* color, the environment it is placed in, orientation, occlusion, etc.).

More concretely, we denote the instance-specific embedding as $v_*$ to describe the instance in $x$ and use it as an adjective (*e.g.,* text prompt 'A photo of a $v_*$ sofa' would generate a specific variant of sofa of $v_*$). To learn the instance-specific embedding $v_*$, we first use the pre-trained text encoder to map prompt 'A photo of a $\mathcal{M}_y$' to get the language embedding $\mathbf{V} = (v_1, \ldots, v_{j-1}, v_j, v_{j+1}, \ldots, v_n)$. Here, $v_j$ denotes the embedding vector that the determiner 'a' maps to. We then modify this vector to insert the instance embedding to create a new vector, $\mathbf{V}_* = (v_1, \ldots, v_{j-1}, v_j, v_*, v_{j+1}, \ldots, v_n)$, where $v_*$ serves as the adjective description. This is similar to adjective token learning in Dream-Booth (Ruiz et al., 2023) and is different from textual inversion (Gal et al., 2022) where the trainable embedding is the noun. We empirically found that treating the learnable embedding as an adjective leads to a faster and more stable training convergence (illustrated in Fig. A9).

The optimization goal of the vanilla textual inversion is given by a modification of Eq. 2, where the only trainable parameter is the embedding $v_*$. However, this is inappropriate for few-shot segmentation because the loss is distributed evenly across the entire image. For training samples where the objects of interest occupy only a small portion of the image, using simple textual inversion results in the generation of images with an unwanted focus on background. To amend this issue, we propose to mask and normalize the loss via,

$$v_* = \underset{v_*}{argmin}\mathbb{E}_{(x,y),\epsilon\sim\mathcal{N}(\mathbf{0},\mathbf{I}),t,\mathbf{v}} \left[ \frac{\sum_{y_i=1}\left\|\epsilon_i - \epsilon_\theta(x_t,t,\mathbf{v})_i\right\|_2^2}{\sum_{y_i=1}1} + \lambda \cdot \frac{\sum_{y_i=0}\left\|\epsilon_i - \epsilon_\theta(x_t,t,\mathbf{v})_i\right\|_2^2}{\sum_{y_i=0}1} \right],$$

(3)

where $i$ is the index of pixels in the image, $y_i = 1$ denotes foreground pixels, $y_i = 0$ denotes background pixels, and $\lambda$ is a hyperparameter used to balance the foreground and background loss. Compared to naïvely masking the loss with foreground masks, Eq. 3 also captures background context such as the surrounding environment and occlusion (an illustration is given in Fig. 2, where the image-conditioned sample shows similar structure to the provided sample). After $v_*$ is optimized, we store $v_*$ to a bank of embedding and sample it along with plain text encodings to generate samples, as illustrated in Fig. 2.

## 3.5 SCENE COMPOSITION VIA COPY-PASTE

As illustrated in Fig. 1, one key difference between synthetic samples from Stable Diffusion and real training samples is the scene layout. Synthetic images are generally *object-centric*, which means a single foreground object occupies most of the image with a clear object-background boundary. In contrast, realistic training samples often have multiple objects per image, with varying sizes of objects and occlusion. Current state-of-the-art method (Wu et al., 2023) attempts to create training samples with more realistic scene layouts by using a heavy combination of image augmentation techniques, which causes artifacts that do not manifest in real samples and is engineeringly demanding.

To address this issue, the insight of AlignDiff is to make use of real samples to condition the synthetic sample generation process and ensure realistic scene layouts. We argue that, with the availability of real samples from base classes, a simple yet effective approach is to compose training samples by copying synthetic objects from Stable Diffusion (Rombach et al., 2022) to real training samples. To meet the storage requirement for practical applications, we follow work in incremental segmentation (Qiu et al., 2023; Cha et al., 2021) and maintain a small subset of base samples $\hat{\mathcal{D}}^B \subseteq \mathcal{D}$ to copy and paste the synthesized samples onto. The subset $\hat{\mathcal{D}}^B$ is constructed in a class-balancing manner such that $\hat{\mathcal{D}}^B$ contains an even number of samples for every base class. During copy-paste, we uniformly select $[1, N_{\max}]$ synthetic samples and randomly place them onto the base images. The details of the implementation can be found in Sec. D.2.

## 3.6 SEMI-SUPERVISED MASK GENERATION

The final step to adapting diffusion models for segmentation training sample synthesis is to generate accurate masks. To exploit the internal representations of Diffusion models for such a purpose, Hertz et al. (2022) proposed to extract image-text cross-attentions and use the response to category names as masks. However, since the pre-training objective of Stable Diffusion (Rombach et al., 2022) does not provide supervision for localizing objects, the attention maps are often imprecise (as illustrated in Fig. A10), necessitating mask refinement and filtering techniques. Most recently, Wu et al. (2023) attempted to address this issue by introducing a noise learning process. Though this approach generates masks of good quality, it requires *full training* of segmentation models for every category, which is expensive and empirically takes 1 GPU day for a few thousand synthetic samples.

AlignDiff builds upon the work of Hertz et al. (2022) and proposes a novel technique for refining coarse masks from diffusion models. We investigate the problem from a novel perspective, where we relate the task setting to semi-supervised learning (Wei et al., 2022). The key insight of AlignDiff is that the expensive noise learning process can be *largely avoided* if we bootstrap the process with a few reference image-mask pairs. Intuitively, we are given two sets initially: $D_{good}$, a set of a few reference image-mask pairs, and $D_{bad}$, a set of many synthetic images with *coarse* masks. The goal is to refine masks in $D_{bad}$ using the high-quality masks from $D_{good}$. This is very similar to the task setting in semi-supervised learning (Wei et al., 2022), in which a widely adopted paradigm is to use knowledge from a small set of good labeled data (comparable to $D_{good}$) to augment the learning on a larger set of data with no label or noisy labels (comparable to $D_{bad}$).

---

**Algorithm 1** Semi-supervised Mask Generation

**Require:** Coarse samples $D_{bad} = \{I_i, M_i\}_{i=1}^N$
**Require:** Given samples $D_{good} = \{I_i, M_i\}_{i=1}^M$
**Require:** FSS model $f_\theta$
**Require:** IoU consensus threshold $\alpha$
  $\theta \leftarrow FSSCond(f, D_{good})$ // Condition FSS
  **for** $i$ from 1 to $N$ **do** // Scoring
      $\hat{M}_i \leftarrow f_\theta(I_i), I_i \in D_{bad}$
      **if** $IoU(\hat{M}_i, M_i) \geq \alpha$ **then**
         $D_{good} \leftarrow D_{good} \cup (I_i, M_i)$
         $D_{bad} \leftarrow D_{bad} \setminus (I_i, M_i)$
      **end if**
  **end for**
  $\theta \leftarrow FSSCond(f, D_{good})$ // Re-condition
  **for** $i$ from 1 to $|D_{bad}|$ **do** // Re-estimate
      $\hat{M}_i \leftarrow f_\theta(I_i), I_i \in D_{bad}$
      $D_{good} \leftarrow D_{good} \cup (I_i, \hat{M}_i)$
  **end for**

---

Inspired by the work of Wei et al. (2022), we design a mask generation process that iteratively migrates samples from $D_{bad}$ to $D_{good}$. The detailed algorithm is given in Algo. 1. More specifically, the algorithm is split into two stages: a scoring stage and a re-estimation stage. We train a few-shot segmentation (FSS) model on the base dataset $\mathcal{D}^B$. During the scoring stage, we condition the

Table 1: AlignDiff can be applied in a model-agnostic fashion to GFSS, where models are required to segment both base and novel classes. AlignDiff improves underlying models across different few-shot settings on COCO-20$^i$ and PASCAL-5$^i$. The best results are **bolded**. HM stands for harmonic mean. *: Simple fine-tuning yields bad performance due to catastrophic forgetting (Cermelli et al., 2021). $^\dagger$: GAPS internally uses copy-paste, which creates realistic scene layouts for GD (Li et al., 2023).

| Method | Base | Novel | HM | Base | Novel | HM |
|---|---|---|---|---|---|---|
| | PASCAL-5$^i$ 1-SHOT | | | PASCAL-5$^i$ 5-SHOT | | |
| PIFS (Cermelli et al., 2021) | 64.1 | 16.9 | 26.7 | 64.5 | 27.5 | 38.6 |
| GFS (Tian et al., 2022) | 65.7 | 15.1 | 24.6 | 66.1 | 22.4 | 33.5 |
| FINETUNE* | 47.2 | 3.9 | 7.2 | 58.7 | 7.7 | 13.6 |
| FINETUNE + GD (Li et al., 2023) | 28.1 | 20.3 | 23.6 | 32.0 | 22.5 | 26.4 |
| FINETUNE+AlignDiff | 66.2(+9.0) | **44.9**(+41.0) | **53.5**(+46.3) | 65.9(+7.2) | 45.1(+37.4) | 53.6(+40.0) |
| GAPS (Qiu et al., 2023) | 66.8 | 23.6 | 34.9 | 68.2 | 43.9 | 53.4 |
| GAPS + GD$^\dagger$ (Li et al., 2023) | 66.8 | 41.2 | 51.0 | **68.5** | 44.0 | 53.6 |
| GAPS+AlignDiff (Qiu et al., 2023) | **67.3**(+0.5) | 43.3(+19.7) | 52.7(+17.8) | 68.4(+0.2) | **47.4**(+3.5) | **56.0**(+2.6) |
| | COCO-20$^i$ 1-SHOT | | | COCO-20$^i$ 5-SHOT | | |
| PIFS (Cermelli et al., 2021) | 40.4 | 10.4 | 16.5 | 41.1 | 18.3 | 25.3 |
| GFS (Tian et al., 2022) | 44.6 | 7.1 | 12.2 | 45.2 | 11.1 | 17.8 |
| FINETUNE* | 38.5 | 4.8 | 8.5 | 39.5 | 11.5 | 17.8 |
| FINETUNE + GD (Li et al., 2023) | 25.8 | 17.2 | 20.6 | 31.9 | 23.6 | 27.2 |
| FINETUNE+AlignDiff | 41.7(+3.2) | 22.4(+17.6) | 29.1(+20.6) | 41.8(+2.3) | 27.9(+16.4) | 33.5(+15.7) |
| GAPS (Qiu et al., 2023) | **46.8** | 12.7 | 20.0 | **49.1** | 25.8 | 33.8 |
| GAPS + GD$^\dagger$ (Li et al., 2023) | 47.1 | 21.8 | 29.9 | 46.5 | 29.0 | 35.7 |
| GAPS+AlignDiff (Qiu et al., 2023) | 46.7(-0.1) | **23.1**(+10.4) | **30.9**(+10.9) | 47.9(-1.2) | **30.3**(+4.5) | **37.1**(+3.3) |

FSS model using the initial $D_{good}$, consisting of a few provided real samples. The conditioned FSS model is then used to predict masks for all samples in $D_{bad}$. If the IoU between the coarse masks and the pseudo annotation predicted by the FSS model exceeds a certain threshold $\alpha$, then AlignDiff deems the coarse masks as high-quality and moves it to $D_{good}$. In the re-estimation stage, AlignDiff reconditions the FSS model using the expanded $D_{good}$ for a more faithful representation. The updated FSS model is then used to generate pseudo labels for all remaining samples in $D_{bad}$. Our semi-supervised mask generation process is much more efficient than the previous noise learning paradigm (Wu et al., 2023), as both the FSS conditioning and inference require no optimization.

# 4 EXPERIMENTS

To demonstrate the efficacy of our method, we apply AlignDiff to two general few-shot segmentation settings. The main results and the ablation study are computed on the challenging Generalized Few-Shot Segmentation **(GFSS)** setting (Tian et al., 2022), which is also known as incremental few-shot segmentation in some literature (Cermelli et al., 2021; Qiu et al., 2023). In addition, to specifically evaluate how our method handles out-of-distribution generation, we apply AlignDiff to Few-Shot Segmentation **(FSS)** on the diverse FSS-1000 (Li et al., 2020) dataset.

## 4.1 EVALUATION SETUP

**Datasets.** For GFSS, we follow previous literature (Qiu et al., 2023; Tian et al., 2022; Cermelli et al., 2021) and use the Pascal-5$^i$ and the COCO-20$^i$ dataset. For FSS, we use the FSS-1000 (Li et al., 2020), a widely used dataset (Tian et al., 2020; Min et al., 2021), to demonstrate how AlignDiff handles out-of-distribution generation. Compared to the Pascal-5$^i$ and the COCO-20$^i$ datasets, which comprise only tens of common objects, FSS-1000 consists of 1,000 classes with uncommon categories.

**Synthesizer Baselines.** We use Grounded Diffusion (GD) (Li et al., 2023) as the main synthesis baseline. Another recent work, DiffuMask (Wu et al., 2023), performs heavy prompt engineering on the Pascal and the COCO dataset, which violates the few-shot setting that the class information is not known beforehand. In addition, DiffuMask requires an expensive noise learning process for every category, which makes comparisons on the COCO dataset impractical. Therefore, we remove the prompt engineering portion of DiffuMask and investigate only the noise learning part for mask generation on a subset of the Pascal-5i dataset in Tab. A2.

Table 2: Results on FSS-1000 (Li et al., 2020) over all 240 testing categories under the 1-shot setting. The OOD (out-of-distribution) classes are determined as classes with negative red bars in Fig. A1. Support set source indicates how the support sets are augmented (*e.g.,* 1R+20S means 1 real sample with 20 synthetic samples). Using simple samples from GD (Li et al., 2023) diminishes the final performance due to OD generation. In contrast, our proposed normalized masked textual inversion (T.Inv.) and semi-supervised masking (S.Mask.) in AlignDiff consistently improve novel IoU, indicating the OOD generation capability of AlignDiff.

| Support Set Source | Synthesis Setup | | Overall IoU | OOD IoU |
| | Conditioning | Mask | | |
| --- | --- | --- | --- | --- |
| 1R | N/A | N/A | 86.5 (Min et al., 2021) | 87.5 |
| 1R+20S | Text | GD (Li et al., 2023) | 81.4 | 80.2 |
| 1R+20S | Text | S.Mask. (**Ours**) | 87.2 | 86.9 |
| 1R+20S | T.Inv. + Text (**Ours**) | S.Mask. (**Ours**) | **88.3** | **88.2** |

**Evaluation Protocol.** For GFSS, we follow existing work (Cermelli et al., 2021; Qiu et al., 2023) and first train the models on $\mathcal{D}^B$ excluding any novel samples. During few-shot learning, $\mathcal{D}^N$ is presented to the model for adaptation. We apply AlignDiff to synthesize 1,000 training samples per novel class to augment $\mathcal{D}^N$ and report metrics on both the base and the novel classes. Note that the evaluation of the validation set is done in a single pass, which is different from the usual episode-based FSS evaluation scheme (Min et al., 2021; Fan et al., 2022). The reported results are averaged across multiple folds in a cross-validating fashion. For each fold, we average results from 5 random runs.

The experiment on FSS follows the standard episode-based protocol (Wang et al., 2019; Min et al., 2021). In both the base training and the few-shot testing stage, the model is presented with episodes that contain a few supporting examples and a query example. The model is tasked to perform binary segmentation of the query sample. In our experiments, we do not modify the base training stage, but we augment the support set $\mathcal{D}^N$ during few-shot testing by supplying extra synthetic samples from AlignDiff, which is conditioned on $\mathcal{D}^N$. Results are average across 1,000 runs.

## 4.2 Main results - Generalized Few-Shot Segmentation (GFSS)

In Tab. 1, we report results on GFSS on the Pascal-5i[i] dataset and the COCO-20[i] datasets. We augment several baselines with AlignDiff to illustrate how much the AlignDiff can help improve the performance of few-shot segmentation in a model-agnostic manner.

**GFSS Baselines.** We use three recent works (Tian et al., 2022; Cermelli et al., 2021; Qiu et al., 2023) from GFSS as baselines. GFS (Tian et al., 2022) proposed the setting of GFSS, but it focuses only on test-time optimization and does not fine-tune on $\mathcal{D}^N$. PIFS (Cermelli et al., 2021) formulates GFSS as a continual few-shot learning task and proposes to fine-tune the model using a few novel samples. Finally, GAPS (Qiu et al., 2023) views GFSS as continual learning and combines memory-replay with copy-paste to further increase the GFSS performance. In this work, we choose GAPS (Qiu et al., 2023) as the main baseline due to its state-of-the-art performance in GFSS.

**AlignDiff consistently improves novel IoU under few-shot settings.** Across all task settings, AlignDiff is able to synthesize more diverse samples to aid few-shot learning of novel categories. Most notably, on the impoverished 1-shot setting, GAPS+AlignDiff improves the novel IoU by approximately 80% on both Pascal-5i[i] and COCO-20[i]. AlignDiff also consistently outperforms GD (Li et al., 2023). Note that the performance gap between AlignDiff and GD is more drastic on simple fine-tuning because GAPS (Qiu et al., 2023) has built-in copy-paste for scene layout. The effect of realistic scene layout is consistent with our findings in the ablation study.

## 4.3 Out-of-Distribution (OOD) Generation for Rare Categories on FSS-1000

Besides introducing more diverse samples to few-shot learning of common categories, AlignDiff also helps with Out-Of-Distribution (OOD) generation of rare categories that plain text conditioning fails. In Tab. 2, we investigate the efficacy of AlignDiff to handle out-of-distribution generation on the FSS-1000 dataset (Li et al., 2020) under the 1-shot setting. We use the HSNet (Min et al., 2021) as

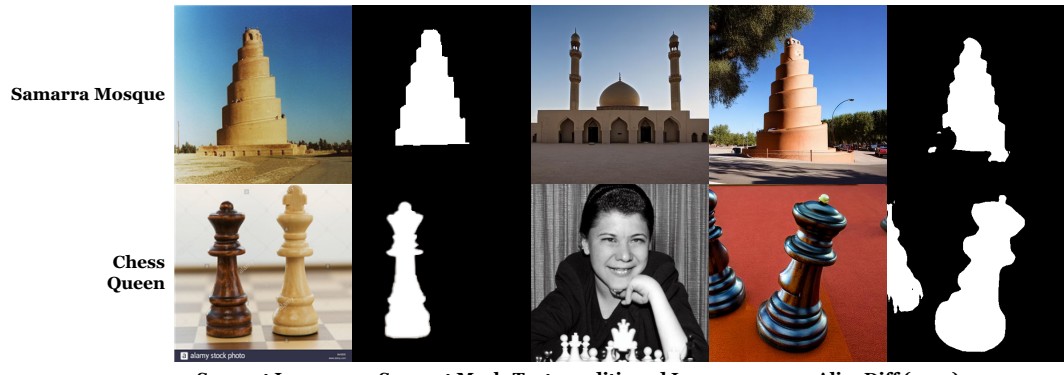

Figure 3: Representative samples for AlignDiff on FSS-1000 (Li et al., 2020). Note how text-to-image synthesis fails for rare classes. AlignDiff generates instances within class and with varying lighting conditions. Note that AlignDiff accurately generates only the 'queen' when the original training image contains another king piece. More samples are included in the supplementary material.

the FSS baseline and apply AlignDiff to provide additional samples for the support set during the few-shot testing. Note that DiffuMask is prohibitively expensive for this scenario since it requires training segmentation models for every category. Thus, we compare only with GD (Li et al., 2023).

Though GD improves performance on Pascal and COCO, it fails drastically on FSS-1000, where the samples it synthesizes negatively impact the final performance. This is due to both inaccurate instances in the images caused by plain text conditioning and inaccurate mask generation. When we use our proposed semi-supervised mask generator, the overall IoU is improved since the masks provided are more accurate. However, the IoU of OOD categories is still worse than conditioning using a single real example even with accurate masks. Finally, we use full AlignDiff and apply the normalized masked textual inversion technique. In this case, both the overall IoU and the IoU of OOD categories surpass the original performance, highlighting the capability of AlignDiff to generate synthetic samples for rare out-of-distribution categories.

For a more in-depth analysis of OOD generation, we refer readers to Sec. C, Fig. A1, and Fig. A2.

**Ablation Study.** We ablate each component in AlignDiff on Pascal-5-3, which is a split of the Pascal-5$^i$ dataset commonly used for ablation. The results are presented in Sec. A and Tab. A1, which validate the effectiveness of all three proposed components in AlignDiff.

**Qualitative samples.** We perform extensive visualization to demonstrate the quality of synthetic samples that AlignDiff generates. Fig. 3 demonstrates the capability of AlignDiff on two OOD categories from FSS-1000, where plain text conditioning fails. Notice that AlignDiff is able to pick up fine-grained details of different types of chess in the synthesized image. More results of FSS-1000 can be found in Fig. A3, and more results of results on Pascal can be found in Fig. A6 to Fig. A8.

**Supplementary materials.** Due to space limit, for more investigations such as the comparison between AlignDiff and the original textual inversion method (Fig. A9) and the comparison to DiffuMask on mask generation (Tab. A2), please refer to the supplementary material.

## 5    CONCLUSION

In this paper, we demonstrate how to adapt large-scale text-to-image synthesis models for few-shot training of segmentation models. AlignDiff generates more realistic training samples by handling out-of-distribution generation, composing training samples with realistic scene layouts, and generating accurate pixel-level annotations. An interesting direction is how to adapt large-scale text-to-image models to directly generate samples with complex scene layouts, which we leave for future research.

**Limitations.** Though AlignDiff can adapt to synthesize instances of rare categories that plain text conditioning fails with as few as a single image, the implicit assumption that AlignDiff holds is that there is no significant domain gap. Therefore, AlignDiff may fail on tasks with drastic domain gaps, such as medical image segmentation.

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

Table A1: Ablating components of AlignDiff on 1-shot Pascal-$5^3$, which is a fold of Pascal-$5^i$. Standard deviations over 5 runs are reported.

| Layout | SMask | T.Inv. | Novel IoU |
|--------|-------|--------|-----------|
| — | — | — | $19.8 \pm 1.6$ |
| ✓ | — | — | $36.8 \pm 1.4$ |
| ✓ | ✓ | — | $38.4 \pm 1.2$ |
| ✓ | ✓ | ✓ | $\mathbf{41.5 \pm 1.2}$ |

Table A2: Comparison of AlignDiff to DiffuMask (Wu et al., 2023) on mask generation on Pascal-$5^3$. GPU hours are reported for the generation of masks for a total of 5 classes with 1,000 images per class on a single RTX3090. AlignDiff is much more efficient with on-par efficiency.

| Method | GPU hours | Novel IoU |
|--------|-----------|-----------|
| AlignDiff (ours) | 1.5 | 41.5 |
| DiffuMask (Wu et al., 2023) | 80 | 40.7 |

## A  ABLATION STUDY

In Tab. A1, we ablate different components in AlignDiff to understand how much each component contributes to the generalized few-shot segmentation performance compared to training directly using synthetic samples from the Stable diffusion (Rombach et al., 2022) model. We use GAPS (Qiu et al., 2023) as the base model to apply AlignDiff to for the ablation study. Following previous works in semantic segmentation (Cha et al., 2021), we use the last five classes of the Pascal VOC dataset as novel classes.

To simulate training directly with pure synthetic samples without any conditioning on real samples, when 'Layout' (which creates samples with diverse scene layouts using copy-paste) is turned off, object-centric images directly from the Stable diffusion model are directly used for training. When 'SMask' (semi-supervised learning for pixel-annotation) is turned off, we use the method of Li et al. (2023) to generate masks in a zero-shot manner. Finally, when T.Inv. (normalized masked textual inversion) is turned off, we generate samples using only class names as text guidance.

**AlignDiff synthesizes samples with complex scene layouts, which is crucial.** We find that copy-pasting dramatically helps with the performance of novel IoU. Compared to training directly with object-centric images from generative models, training with images composed via copy-paste nearly doubles the novel IoU on the Pascal-$5^i$ dataset, which validates the importance of realistic scene layout of training samples for segmentation models.

**AlignDiff provides high-quality pixel-level annotations.** Compared to the zero-shot segmentation method proposed by Li et al. (2023), which requires a well-trained COCO instance segmentation model to provide annotation during base training, our semi-supervised mask generation method provides pixel-level annotations of higher quality and improve the novel IoU by approximately 10%.

**Textual inversion generates more diverse samples.** Compare to samples synthesized with pure text conditioning, the instance-specific embedding learned by AlignDiff introduces more diversity and further improves the novel IoU by approximately 10%.

## B  DATASET DESCRIPTION

Pascal-$5^i$ is artificially built from the PASCAL VOC 2012 dataset (Everingham et al., 2010) with augmented annotations from the SBD (Hariharan et al., 2011) dataset. The original VOC segmentation dataset provides segmentation annotations for 20 object categories, and the Pascal-$5^i$ dataset manually splits the original dataset into 4 folds for cross-validation. For each fold, 5 categories are selected as novel categories, while the remaining 15 are regarded as base categories. The construction of the COCO-$20^i$ dataset uses the 80 thing classes in COCO similarly, where the dataset is split into 4 folds with 20 categories per fold.

## C  MORE RESULTS ON FEW-SHOT SEGMENTATION FOR OUT-OF-DISTRIBUTION GENERATION

For a more in-depth analysis, we present a class-wise IoU difference between the second row and the first row of Table. 2 in Fig. A1. We can observe that the effects of synthetic samples generated by GD are mixed for individual classes. For 50 out of the 240 categories in the testing split of FSS-1000,

synthetic samples generated by GD improve the final IoU. However, for the rest 190 categories, GD-generated samples negatively impact the IoU, which suggests that the generated image-mask pairs for these classes are inaccurate. We mark these 190 categories as 'out-of-distribution' categories since synthetic samples generated by GD negatively impact the final segmentation performance. The results presented in Table 3 of the main paper are computed using the top-5 classes at the end ('pidan', 'Samarra Mosque', 'Chess queen', 'American Chamelon', and 'Phonograph'). Analysis of different failure patterns for out-of-distribution categories is given in Sec. E and Fig. A3.

Finally, AlignDiff, which conditions both the image generation process and the mask generation process using a few provided real images, addresses both aforementioned issues. This results in improved IoU compared with all other settings using a single real sample and using text-conditioned synthetic samples. The class-wise detailed analysis of the IoU difference between AlignDiff-augmented few-shot segmentation and 1-shot real-sample-only few-shot segmentation is given in Fig. A2. Synthetic samples generated by AlignDiff improve the IoU of 178 out of the 240 testing categories, which results in better overall IoU and the OOD IoU of classes where GD-synthesized samples negatively impact the segmentation performance.

## D   IMPLEMENTATION DETAILS

### D.1   TEXT-TO-IMAGE SYNTHESIS MODEL

We use the pre-trained checkpoints and the codebase of Stable Diffusion Rombach et al. (2022), which is available at[1]. The weights of the Stable Diffusion are frozen.

### D.2   COPY-PASTE FOR SCENE LAYOUT

During copy-paste, we randomly select a generated sample and paste the selected object onto an image. Since previous work Ghiasi et al. (2021) has found that simple copy-paste serves as a good image augmentation technique without blending or harmonization technique, we follow the same practice for simplicity. We randomly scale the object and randomly translate the object placement. The number of objects is uniformly select from $\{1, 2, 3\}$, with scaling factors uniformly sampled from $[0.1, 2.0]$. No rotation or color jittering is used.

### D.3   TRAINING DETAILS FOR GENERALIZED FEW-SHOT SEMANTIC SEGMENTATION

We follow the standard sequential learning procedure that two recently published works, PIFS Cermelli et al. (2021) and GAPS Qiu et al. (2023), use in their papers. Specifically, these two works separate the learning process into two stages. In the first base learning stage, they use the standard training procedure to train a segmentation model on the base datasets. To avoid information leaks from unseen classes, the base dataset excludes all images with at least one pixel of the novel classes. The architecture of the segmentation model is DeepLab-V3 Chen et al. (2017), which uses the ResNet-101 backbone and replaces the last per-pixel classification layer with a cosine-similarity-based per-pixel classification layer. Following GAPS Qiu et al. (2023), on both the COCO and the PASCAL VOC datasets, the base training uses a batch size of 32, a polynomial learning rate schedule with an initial learning rate of 0.01, and training for 20 epochs. The SGD optimizer is used with 0.9 momentum and 0.0001 weight decay. Following existing work Cermelli et al. (2021); Qiu et al. (2023), standard augmentation techniques such as random scaling, random cropping, and random horizontal flipping are used.

During the novel learning stage, we again resort to the training settings proposed by GAPS Qiu et al. (2023). We fine-tune the segmentation models every time the model is presented with a new class (*e.g.,* a total of five fine-tunings for 5 classes on each split of the Pascal-5$^i$ dataset). For GAPS-based methods, on the PASCAL VOC dataset, we train the segmentation model for 200 iterations using a batch size of 16, a backbone learning rate of 0.001, and a classifier learning rate of 0.01. For the GAPS-based method, we use the memory-replay buffer construction strategy proposed in GAPS Qiu et al. (2023) to construct a small subset of 500 base examples, $\hat{\mathcal{D}}^B$, for copy-pasting the novel samples onto. For finetune-based methods, the training settings such as batch size are similar to

---

[1] https://github.com/CompVis/stable-diffusion

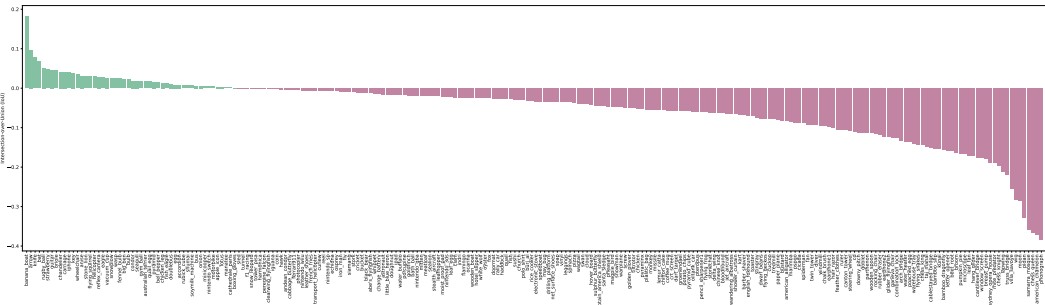

Figure A1: GD Li et al. (2023) may synthesize samples that are harmful to the overall IoU. Class-wise IoU difference of HSNet Min et al. (2021) on the FSS-1000 dataset Li et al. (2020) under the 1-shot setting with support set of 1) only 1 real sample and 2) 1 real sample and 20 text-conditioned synthetic samples generated by GD Li et al. (2023). Green bars denote classes whose text-conditioned samples improve the final IoU, whereas red bars denote classes whose text-conditioned samples negatively influence the final IoU. We mark classes with red bars as out-of-distribution categories. 190 out of 240 total categories are considered out-of-distribution because synthetic samples generated by GD are harmful to the segmentation performance.

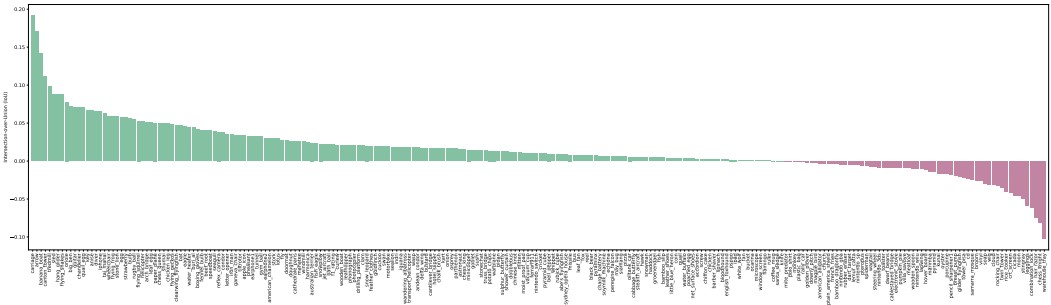

Figure A2: AlignDiff handles out-of-distribution generation and improves the overall IoU of 1-shot segmentation on the FSS-1000 dataset Li et al. (2020). Class-wise IoU difference of HSNet Min et al. (2021) on the FSS-1000 dataset Li et al. (2020) under the 1-shot setting with support set of 1) only 1 real sample and 2) 1 real sample and 20 synthetic samples generated by AlignDiff (ours). Green bars denote classes whose synthetic samples improve the final IoU, whereas red bars denote classes whose synthetic samples negatively influence the final IoU. AlignDiff improves IoU for a total of 178 out of 240 classes and improves the overall IoU.

GAPS. However, the subset of base examples to paste on, $\hat{\mathcal{D}}^B$, is constructed by randomly selecting 500 examples from the base dataset.

In addition to the learning settings, we also list details of how AlignDiff uses the synthetic samples. When AlignDiff is combined with plain fine-tuning, we use the copy-paste strategy to create samples with realistic scene layouts. When it is combined with GAPS Qiu et al. (2023), which is a method based on copy-paste, we follow the procedure described in Qiu et al. (2023) and directly add synthetic samples with their masks as candidates for copy-paste in GAPS. In both cases, 50% of the synthetic samples are selected from the few-shot samples or samples conditioned via normalized masked textual inversion; whereas the images of the rest of the 50% samples are purely synthetic using text conditioning.

## D.4 TRAINING DETAILS FOR FEW-SHOT SEMANTIC SEGMENTATION

We follow the standard training procedure in few-shot semantic segmentation using support-query episodes Wang et al. (2019); Min et al. (2021); Fan et al. (2022). In particular, we use HSNet Min et al. (2021), which is a commonly used method in few-shot segmentation, to investigate the performance of AlignDiff to handle out-of-distribution generation on the FSS-1000 dataset. We use the same

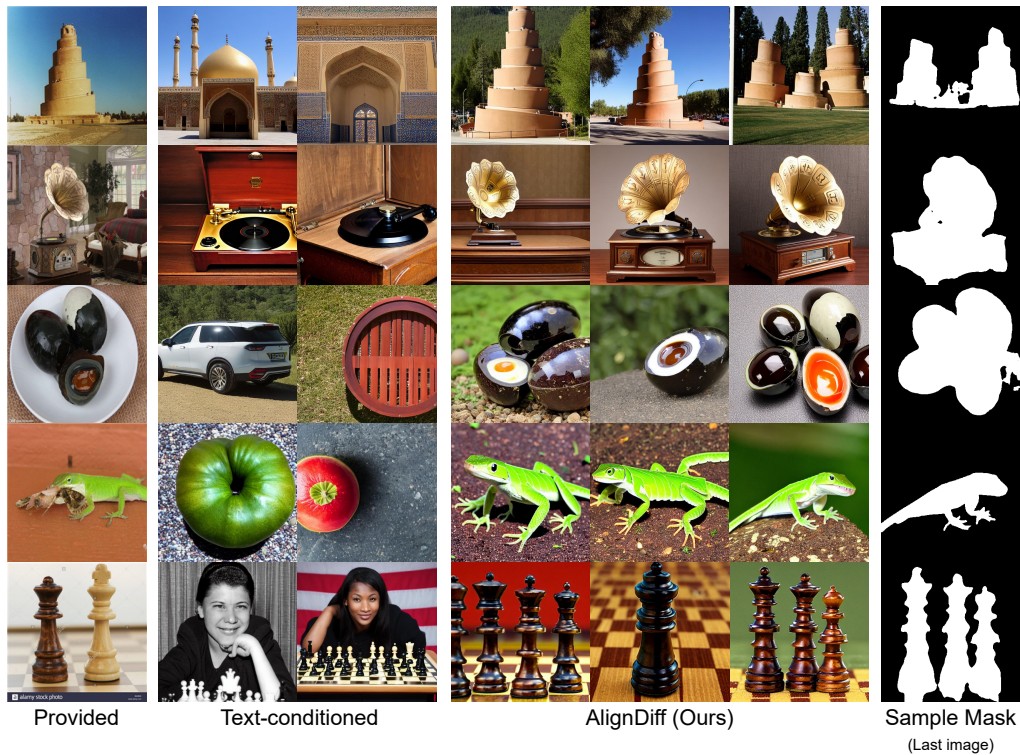

Figure A3: More qualitative results of our Normalized Masked Textual Inversion v.s. plain text conditioning on the FSS-1000 Li et al. (2020) dataset. The 5 categories (Samarra Mosque, phonograph, Pidan, American Chameleon, and chess queen) are the 5 rightmost 'out-of-distribution' categories illustrated in Fig. A1. AlignDiff successfully augments the training process of these 5 categories that GD fails.

training settings in HSNet, where support and query images are resized to 400 by 400, and parameters are optimized via an Adam optimizer with a learning rate of 1e-3.

Unlike PIFS Cermelli et al. (2021) and GAPS Qiu et al. (2023) from generalized few-shot segmentation, HSNet does not require fine-tuning on novel data. Instead, given a support set and a query image, HSNet refines the predictions of the mask of the query image using the samples given in the support set. Therefore, we focus on augmenting the support set with additional synthetic samples to improve the performance. Since the FSS-1000 dataset contains mostly object-centric samples, we do not use the copy-paste scene composition module in AlignDiff. In order to avoid including degenerate samples for out-of-distribution categories, we perform a simple estimation step. For every text-conditioned sample, we use it as a 1-shot support set and treat the given real support image as a query image to perform 1-shot segmentation. If the IoU of the predicted mask and the given query mask exceeds a pre-defined threshold $\beta$ (we set $\beta = 0.5$), then this particular text-conditioned sample is added to the final support set.

## E ANALYSIS OF FAILURE PATTERNS OF PLAIN TEXT CONDITIONING

As illustrated in the text-conditioned sample column of five out-of-distribution categories from FSS-1000 Li et al. (2020) in Fig. A3, a few types of failure patterns can be observed for GD Li et al. (2023) and Stable Diffusion Rombach et al. (2022) in general when the generative process is guided using plain texts. These factors necessitate methods for conditioning the generative process using a few input image-mask pairs.

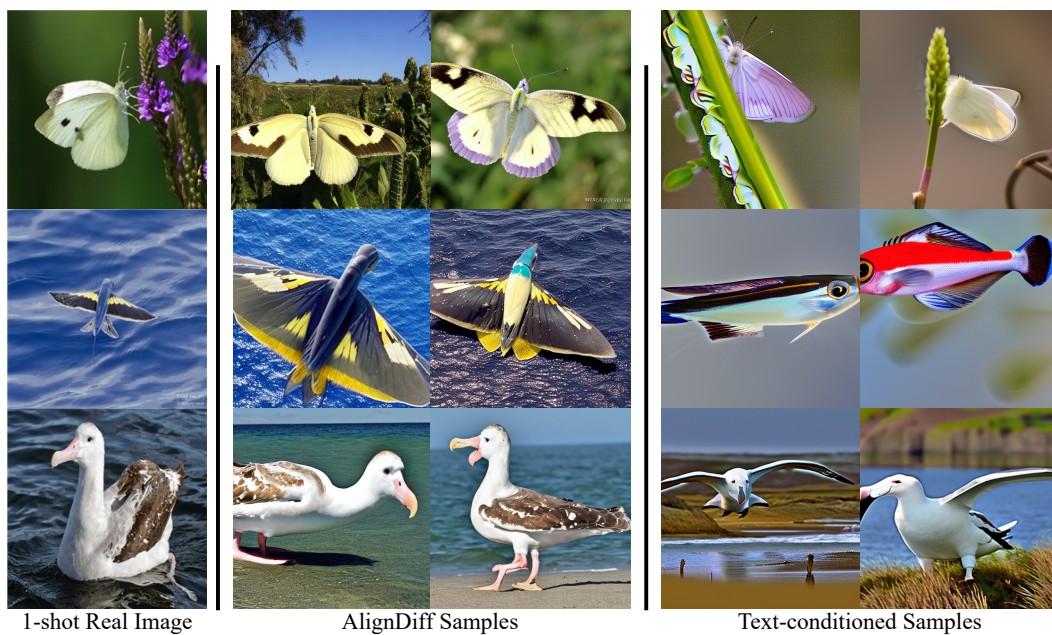

Figure A4: Qualitative results on the validation split of the FSS-1000 dataset. Notice how AlignDiff is able to generate samples for uncommon categories. Category names from top to below are: *Cabbage butterfly, clearwing flyingfish, and wandering albatross*. Notice that AlignDiff is able to capture fine-grained details of objects such as the black spots on the cabbage butterfly and texture of albatross.

- **Inaccurate attention.** Stable diffusion may attend to only a single word in a multi-word object phrase, which is shown in the 'samarra mosque' sample in the first row. Stable Diffusion incorrectly attends to only the word 'mosque' and generates images of a common mosque, rather than the Samarra mosque. The chess queen samples in the fifth row fail in a similar pattern, where Stable Diffusion incorrectly attends mainly to the word 'queen'.

- **Uncomprehensive description.** The second row shows samples for the 'phonograph' class from FSS-1000. In this case, images of phonograph in FSS-1000 are all instances of vintage phonographs with copper horns. However, when given the prompt 'a photo of a phonograph,' Stable Diffusion generates images of common phonographs with no horns and a turnable player, as shown in the samples.

- **Rare concept.** Pidan (third row) and American Chameleon (fourth row), as a type of uncommon Asian traditional food, are also classes from the FSS-1000 dataset. However, Stable Diffusion fails completely on those novel concepts when given only text prompts and generates irrelevant images.

## F   MORE QUALITATIVE RESULTS

We provide additional representative synthetic samples generated by our AlignDiff and compare them with GD Li et al. (2023) in this section.

Fig. A5 gives results of text-conditioned image synthesis for the last five classes of the PASCAL VOC dataset with masks generated by GD Li et al. (2023). We notice that the PASCAL VOC dataset contains only commonly seen classes. Thus, Stable Diffusion can generate diverse instances for PASCAL VOC using only text conditioning. However, GD fails to generate accurate masks, which hinders the training of segmentation models.

Fig. A6 shows results of text-conditioned image synthesis with masks generated by the semi-supervised mask generator that we proposed. We can notice how AlignDiff is able to generate masks with crispy boundaries.

| Potted Plant | Sheep | Sofa | Train | TV Monitor |
|---|---|---|---|---|

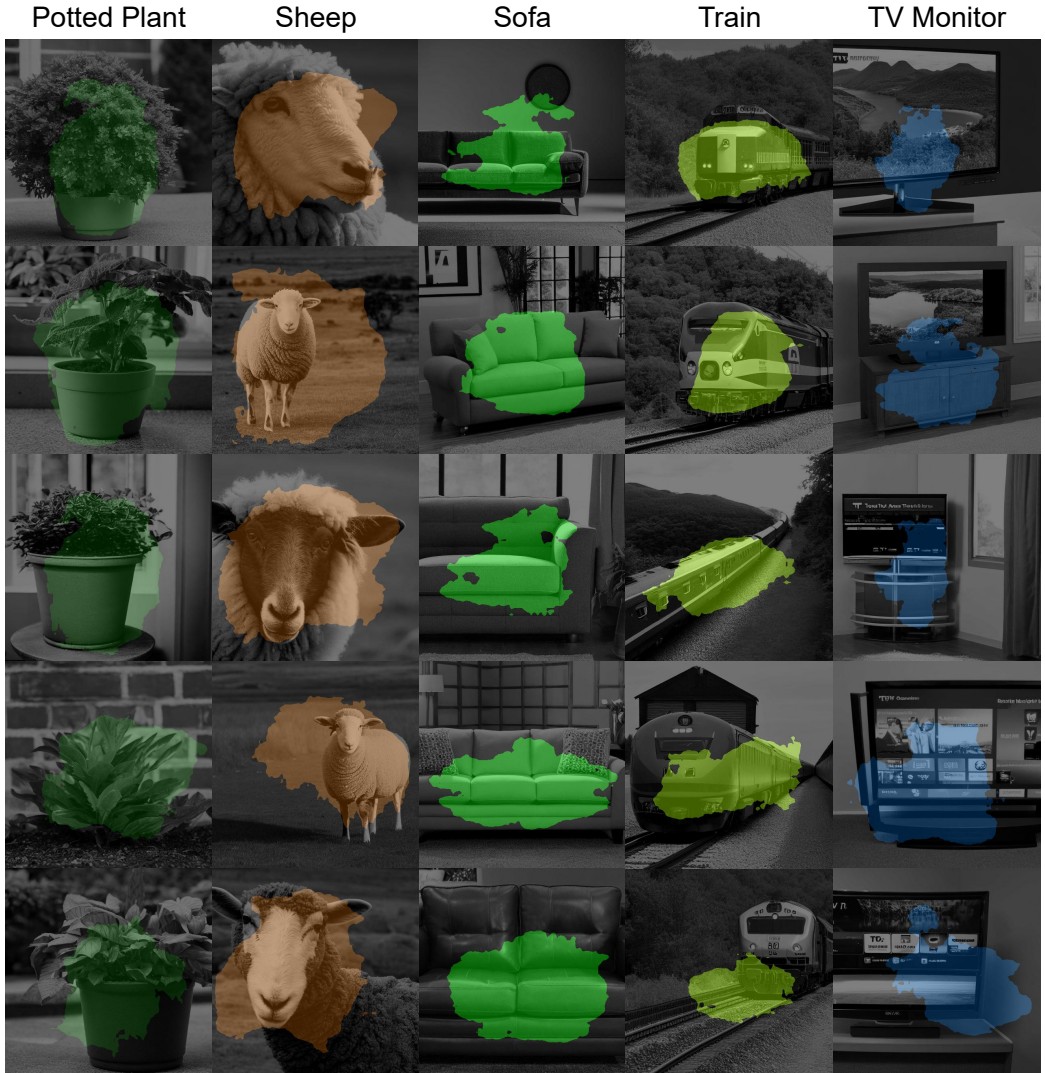

Figure A5: Qualitative results on the PASCAL-5-3 split using plain text conditioning. Masks are generated by GD Li et al. (2023). Best viewed in color.

Fig. A7 shows interesting qualitative results on the PASCAL-5-3 split using the proposed normalized masked textual inversion method for conditioning. The first row contains the 1-shot image-mask pairs provided to AlignDiff, and the rest of the image-mask pairs are generated by AlignDiff. The figure illustrates two intriguing findings: 1) AlignDiff is able to generate relevant images despite the objects of interest may only occupy a small region in the original image, which is not possible with plain textual inversion Gal et al. (2022) and 2) Compared to text-conditioned samples in Fig. A5 and Fig. A6, samples synthesized by AlignDiff increas the diversity of synthesized samples (*e.g.,* AlignDiff generates sofas with more realistic camera viewpoint and CRT monitors with variations). This helps better capture intra-class variation, which is also quantitatively validated in the ablation study in the main paper.

Potted Plant  Sheep  Sofa  Train  TV Monitor

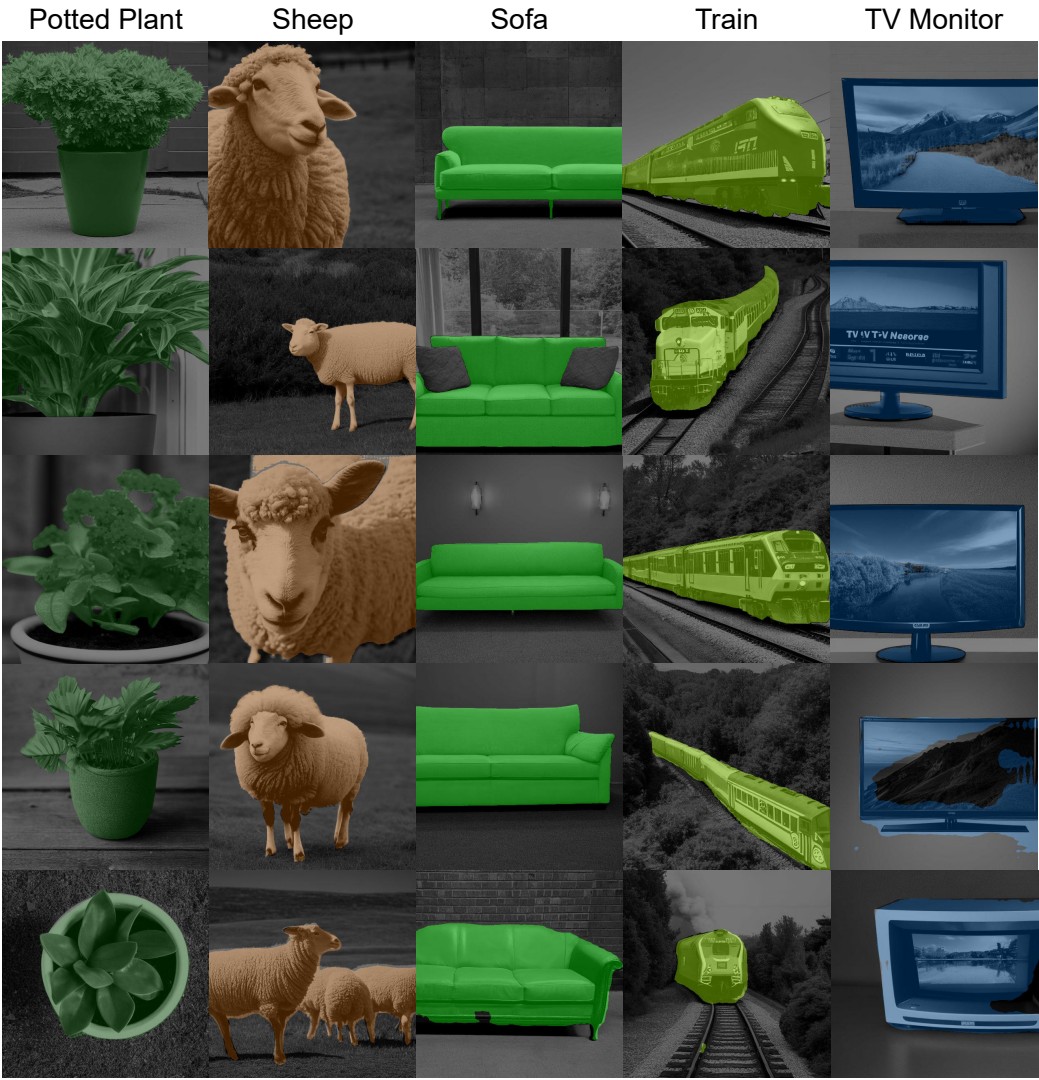

Figure A6: Qualitative results on the PASCAL-5-3 split using plain text conditioning. Masks are generated by our AlignDiff. Best viewed in color.

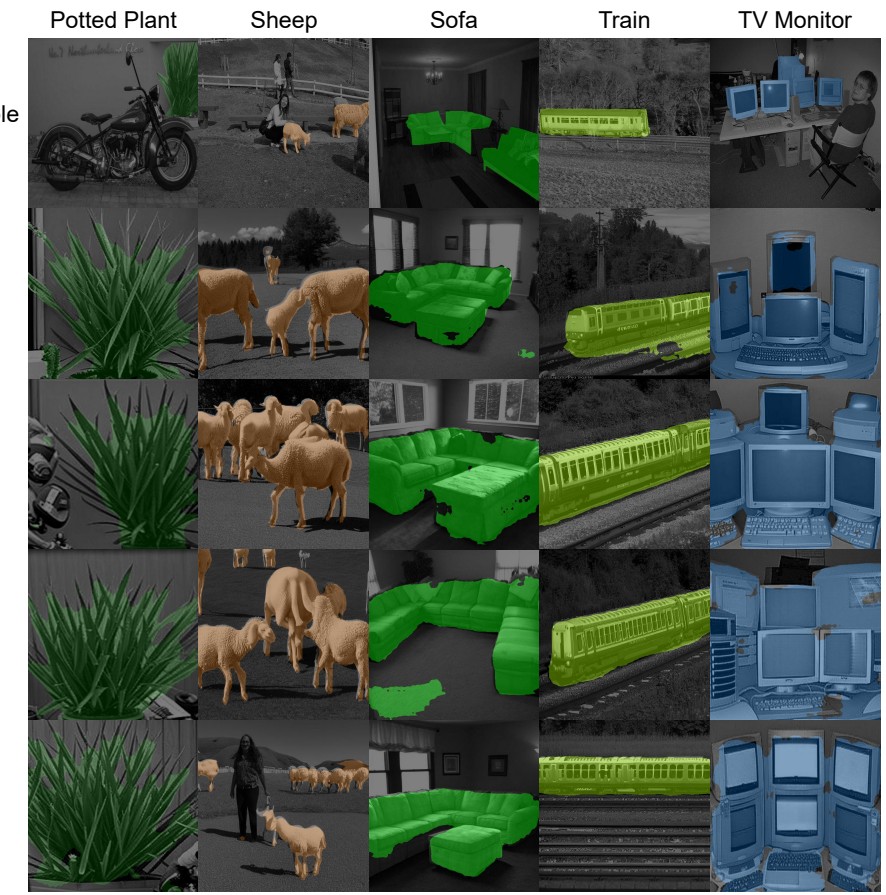

Figure A7: Qualitative results on the PASCAL-5-3 split using the proposed normalized masked textual inversion method for conditioning. The first row contains the 1-shot image-mask pairs provided to AlignDiff, and the rest of the image-mask pairs are generated by AlignDiff. The figure illustrates two interesting findings: 1) AlignDiff is able to generate relevant images despite the objects of interest may only occupy a small region in the original image, which is not possible with plain textual inversion Gal et al. (2022) and 2) Compared to text-conditioned samples in Fig. A5 and Fig. A6, synthesized samples here increase the diversity of training samples (*e.g.,* AlignDiff generates sofas with more realistic viewport and CRT monitors with variations). This helps better capture in-class variation, which is also quantitatively validated in the ablation study in the main paper. Best viewed in color.

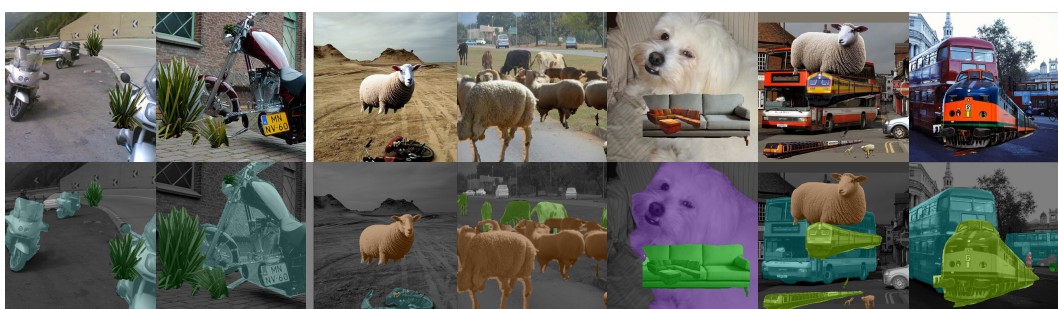

Figure A8: Training samples synthesized for the Pascal-5-3 dataset split generated by AlignDiff. Note the complex scene layout composed by copy-paste and the accurate masks of synthesized samples from semi-supervised mask generation.

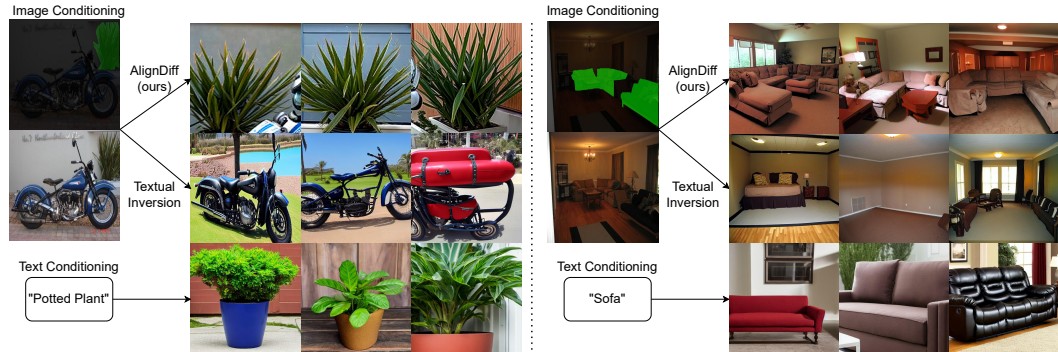

Figure A9: Comparison between our proposed Normalized Masked Textual Inversion in AlignDiff, Naïve Textual Inversion, and plain text conditioning. Text-conditioned samples exhibit variance in texture and lighting conditioning, but lack geometric variations and realistic scene layout. Naïve textual inversion captures the realistic scene layout, but often leads to undesired attention of unrelated objects such as the 'motorcycle' in the left-hand side images and 'room floor' in the right-hand side images when the novel object occupies only a small portion of the image. Note how our proposed Normalized Masked Textual Inversion can successfully generates synthetic samples in such scenario.

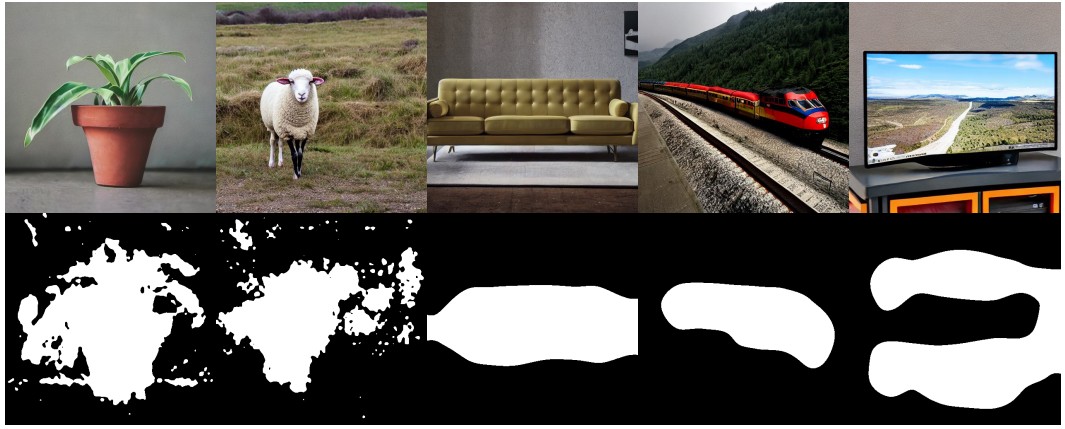

Figure A10: Coarse masks generated by exploiting cross-attention of Stable Diffusion (Hertz et al., 2022; Wu et al., 2023).

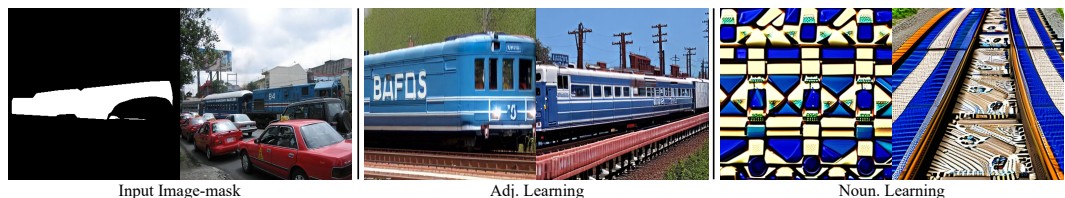

Input Image-mask       Adj. Learning       Noun. Learning

Figure A11: Qualitative samples of optimizing adjective embedding v.s. noun cembedding. When the object of interest occupies only a small region in the input image, noun serves as an additional regularizer that retains the semantic concept.

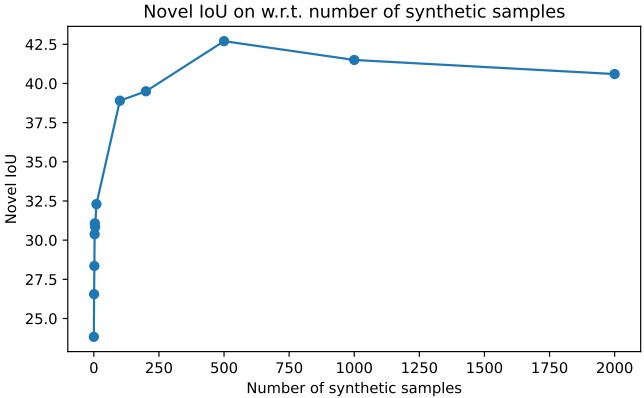

Figure A12: Relationship between the number of synthesized images and novel IoU on Pascal-5-3 under the 1-shot setting. We can ob- serve that the performance saturates at around 500 synthesized samples. To avoid overfitting to a specific task setting, we use 1,000 synthesized samples throughout all experiments in our pa- per.

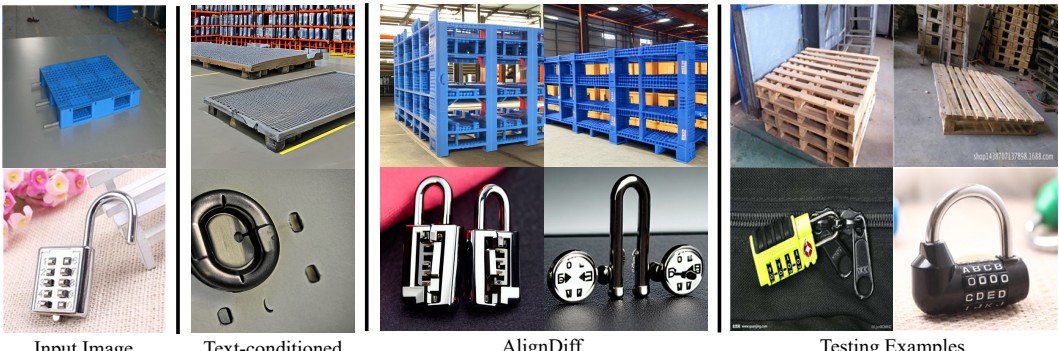

| Input Image | Text-conditioned | AlignDiff | Testing Examples |

Figure A13: Qualitative samples of categories where AlignDiff augmentation fail (correspond to red categories in Fig. A2). The augmentation fails if both text-conditioned samples and AlignDiff-generated samples have gaps from the testing distribution. For instance, the first row demonstrates 'warehouse tray', where AlignDiff is given a blue tray and the testing distribution is made of wooden trays. Similarly, the texture and style of combination locks in the second row varies significantly on a sample-by-sample basis.

