# OpenReview forum: "AlignDiff: Aligning Diffusion Models for General Few-Shot Segmentation"
_ICLR.cc/2024/Conference — Submitted to ICLR 2024_

### Official Review · Reviewer_cYiM · 2023-10-19

**Soundness:** 3 good
**Presentation:** 1 poor
**Contribution:** 2 fair
**Rating:** 5
**Confidence:** 4

**Summary:**

This paper aims to utilize pre-trained text-to-image diffusion models for few-shot segmentation. It points out three levels of misalignments that arise when utilizing pre-trained diffusion models in segmentation tasks: 1) text prompt may not generate desired instances; 2) may fail on multi-object scenes; 3) diffusion models cannot generate segmentation masks.

To solve 1), it binds an instance specific word embedding with the given real examples. To solve 2), it combines
synthesized instances with real images to generate training samples with more realistic layouts. To solve 3), it use semi-supervised learning (Wei et al., 2022) and condition the generative process on provided novel samples.

The experiments are done on Pascal-5, COCO-20, and FSS-1000. Compared to previous methods, the proposed method achieves the SoTA.

**Strengths:**

The proposed method seem robust as it achieves on-par performance even if it is combined with simple fune-tuning, while other methods suffer from significant decrease in performance.

**Weaknesses:**

1. As few-shot segmentation might refer to semi-supervised segmentation with very few annotated examples and lots of unannotated examples in some literature, different from the setting in this paper, I suggest to state the problem setting in the very beginning of the paper.
2. If I recall correctly, using a special adjective token for a specific instance was proposed in [A], but it seems claimed as one of the contribution of this paper.
3. This paper does not clearly explain the "copy-paste" process, but simply cites another paper. An example figure would be nice. The images in Figure 2 is too dark to see clearly. If the space is not enough, I suggest to remove the introduction of diffusion models as it is becoming a common sense in this area.
4. The proposed method sounds very expensive. For every category, users need to personalize Stable Diffusion with text-inversion first, and then train the segmentor over and over again to contain the new generated masks into training set. However, the paper seems to only compare the mask generation speed. I would suggest to compare the whole process speed to benefit the community.
5. The main performance gain seems to come from combining two papers: [B] and [C], which correspond to the contribution 2) and 3) in the introduction, respectively.

[A] DreamBooth: Fine Tuning Text-to-Image Diffusion Models for Subject-Driven Generation

[B] GAPS: Few-shot incremental semantic segmentation via guided copy-paste synthesis

[C] An embarrassingly simple approach to semi-supervised few-shot learning.

**Questions:**

1. Related to weakness 3), does the "copy-paste" process requires post-process harmonious method? If not, how to make sure the generated image make sense to the layout? Does it harm the performance?
2. Related to 1), with only a few examples of the new instance, how to obtain multi-object layout? Does the layout include the new instance?

---

> ### Author Response · Authors · 2023-11-21
> **Response to reviewer cYiM**
>
> We thank the reviewer for the valuable comments. Below we address all the concerns one by one.
>
> ## Clarification of task setting
>
> We thank the reviewer for pointing out the possible confusion of our semi-supervised mask generation method with semi-supervised segmentation. We have revised the introduction section so that the method and task settings are clearly stated.
>
> ## Clarification of contribution
>
> Though both AlignDiff and Dreambooth propose to learn instance-specific embeddings, AlignDiff goes a step beyond DreamBooth to adapt Diffusion models for learning instance-specific embeddings to generate training samples with a revised loss scheme. Specifically, the normalized masked textual inversion loss in AlignDiff improves from DreamBooth in two major ways: 1) Dreambooth requires tens of images of the target object, whereas AlignDiff works with as few as one image of the target category and 2) Dreambooth requires up-close images of the objects where the desired object takes up most of the foreground. In comparison, AlignDiff is able to work with small objects. (For qualitative samples of how AlignDiff generates images of small objects compared to existing methods such as DreamBooth or Textual Inversion, refer to Fig. A9 in the supplementary).
>
> ## Clarification of copy-paste implementation
>
> We revised the manuscript to explain more details and exact numbers in Section D.2 and refer readers to this section in the method section. In addition, we provide visualized copy-and-pasted images in Fig. A8 in the supplementary material.
>
> Note that we intentionally implement the copy-paste in a simple manner because investigating the details of copy-paste is not a goal of this paper. Our goal is to argue that previous work (i.e., DiffuMask) uses a heavy image augmentation technique to generate samples with more realistic layouts for training because they overlook a simple yet effective method - copy-and-paste - to combine generated data with real data.
>
> The main goal of the preliminary is to relate our revised loss scheme in Eq. 3 with Eq. 2 from the original diffusion work.
>
> ## Timing of the method
>
> Timing is measured using a single RTX3090 GPU and with 512 x 512 resolution. AlignDiff operates in two stages. In the first stage, AlignDiff takes ~5 minutes to inverse the embedding for few-shot samples.
>
> In the second stage of sample generation, AlignDiff performs
>
> Diffusion generation: ~4s per image
> Coarse mask: <0.1s per image
> Semi-supervised refinement: <0.15s per image
>
> We believe that AlignDiff is quite suitable for practical applications with its two-step scheme. The cost of the initial sample generation is amortized over the generation process.
>
> ## Explanation of performance gain
>
> We respectfully disagree with the reviewer that the performance gain comes from combining papers [B] and [C].
>
> **Relationship with [B]**: though AlignDiff and [B] both use copy-paste [R1], the novelty of our approach does not lie in the fact that we use copy-paste. Instead, we argue the complexity of the approach in previous work such as DiffuMask, where heavy image augmentation techniques are used, is because they overlook real data. Copy-paste is merely an effective solution to relate generated data to real data.
>
> **Relationship with [C]**: our mask generation method has completely no technical similarity with [C]. The one and only reason we mention [C] in our paper is to establish a graceful similarity overlooked by previous work between mask generation and semi-supervised learning. [C] is designed for semi-supervised classification, whereas our method focuses on generating masks for images generated by Stable Diffusion. We refer to [C] only to mention its task setting; whereas the technical contributions from [C] are irrelevant to our approach.
>
> ## Harmonization and clarification of object layout in copy-paste
>
> Previous work in copy-paste [R1] has found that copy-pasting without using any blending technique shows similar performance with implementations that use blending to harmonize the generated images. To make consistent with these previous works, we also did not implement harmonization technique in our paper and similarly found that even generated images with uncommon layouts yields significant performance gain (example generated images can be found in Fig. A8). An interesting direction would be to further harmonize the image to ensure the consistency of semantic layout, but the focus of this work is to identify the misalignment of diffusion models for generating training examples, so we leave this to future work.
>
> For more technical details, the current implementation obtains multi-object layout by randomly translating and randomly scaling objects before copy-paste. The instances that are used for copy-pasting include both real samples and samples generated from Stable Diffusion.
>
> ## References
>
> [R1] Ghiasi, Golnaz, et al. "Simple copy-paste is a strong data augmentation method for instance segmentation." CVPR. 2021.

---

> > ### Comment · Reviewer_cYiM · 2023-11-23
> > **Official response to the authors**
> >
> > Thank you for the clarification. My concerns are partially addressed, but my main concern remains: the contribution is limited. For example, DreamBooth proposed to associate the new object with an adjective word, but it is not even cited in the current manuscript version. Besides, DreamBooth typically works when tuning on 4-6 images instead of tens.
> > Overall, I do believe there is contribution in this paper, but it is not claimed correctly. Therefore, I keep my original rating.

---

> ### Author Response · Authors · 2023-11-23
> **Follow-up Response to reviewer cYiM**
>
> Dear Reviewer cYiM,
>
> Thank you for your reply and further clarification. We unintentionally made an honest mistake in our previous response – we apologize for this. Instead of tens of images, DreamBooth [A] indeed does work with as few as 4-6 images with the desired objects taking up most of the pixels. However, it is still inappropriate for generating data for few-shot segmentation, where the model is required to work with as few as a single image. In contrast, our AlignDiff works with as few as a single image, as shown in Fig. A4 in the appendix.
>
> We have also updated our paper accordingly to cite DreamBooth. Initially, our method was motivated by and more related work textual inversion, which is a predecessor to DreamBooth that also performs image-to-text inversion. Note that we did not intend to claim learning tokens for adjectives as a part of the contributions for the paper. Instead, we discussed it as a detailed implementation and for best reproducibility. Our main contribution of this part is the revised loss term (which we term normalized masked textual inversion). This revised loss term is what allows us to learn with as few as a single image.
>
> We attached the revised paragraph in the updated paper PDF below for your convenience.
>
> > ... serves as the adjective description. This is similar to adjective token learning in DreamBooth (Ruiz et al., 2023) and is different from textual inversion (Gal et al., 2022) where the trainable embedding is the noun.
>
> We hope this clarifies your concern. We are happy to discuss more if you have any remaining questions.

---

### Official Review · Reviewer_ZQEG · 2023-11-01

**Soundness:** 3 good
**Presentation:** 3 good
**Contribution:** 2 fair
**Rating:** 3
**Confidence:** 4

**Summary:**

The paper presents a data augmentation approach for few-shot image segmentation. The aim is to synthesise training samples of the novel object categories that are segmented from the few-shot samples. The approach is based on proposing three modifications to the text-to-image stable diffusion for image generation. First, the image sample generation is based on creating a bank of banks of embeddings with the proposed mask and normalising the loss related to the textual inversion (Gal et al 2022). Second, more training samples are generated in a copy-paste manner by including objects generated with stable diffusion in the available training data. Third, a few-shot segmentation (FSS) model is trained with a set of appropriate image-segmentation pairs.  The FSS model is used to find more adequate pairs and is trained again with a larger pool of samples. The approach shows reasonable performance on several standard benchmarks for few-shot segmentation.

**Strengths:**

+ The paper is well written and easy to follow. In addition, the related work is complete and discussed in detail. The method is also well presented.

+ The method provides solid results for almost all evaluations.

+ The ideas proposed are easy to implement in any latent diffusion model.

+ The paper addresses an open problem for segmentation-like tasks. It is challenging to generate scenes with pixel-level masks using diffusion models.

**Weaknesses:**

- (Major) The three main contributions of the paper are extensions of existing approaches. This is not a problem, but in all cases the new approach is minor. For example, the copy-paste idea is used out of the box. Similarly, the iterative training of the few-shot segmentation model does not contain any particular innovation. The paper has limited novelty.

- (Major) The 1-shot results would be helpful as they are also common to the previous work.  In addition, a comparison with recent approaches would be important. For example, Xu, Qianxiong, et al. "Self-Calibrated Cross Attention Network for Few-Shot Segmentation". Proceedings of the IEEE/CVF International Conference on Computer Vision. 2023. It would be useful to discuss when the paper does not achieve SOTA results and why. Overall, the results show that the proposed ideas do not show much improvement.

- The term "synthetic distribution" is a bit confusing because it is usually associated with the generation of data from a simulator. This is not the case for the problem under consideration. Generated / realistic data distribution would be more appropriate.

- The term "out-of-distribution generation" is also confusing. There is no discussion of what is in-distribution information in terms of prompts or generated images. The paper may refer to the additional image variations given a prompt as OOD. This is not clear. However, OOD here differs from the common use of it in uncertainty estimation.

**Questions:**

- It would be interesting to discuss whether the method is limited to latent diffusion models or generalisable to more diffusion approaches.

---

> ### Author Response · Authors · 2023-11-21
> **Response to reviewer ZQEG**
>
> We thank the reviewer for the valuable comments. Below we address all the concerns one-by-one.
>
> ## Novelty
>
> We respectfully disagree with the reviewer that our work has limited novelty. In particular, we would like to highlight our novelty from two aspects:
>
> (1) **Paradigm-level novelty:** We investigate a novel and interesting task setting in this paper - adapting text-to-image diffusion models to generate training samples (as recognized by reviewers ofKT and bS35). We went beyond naively using generated samples to facilitate the training process. Instead, we identify issues in existing works (e.g., mask generation efficiency and quality) and issues that are overlooked by existing works (e.g., generation for rare categories).
>
> (2) **Methodology-level novelty:** Previous work, such as DiffuMask, generates image-mask pairs with expensive GPU hours. In contrast, AlignDiff handles both generation for rare categories and generates high-quality masks with great efficiency, which is not possible with previous methods and has great potential for practical applications. We believe this constitutes technical novelty of our method.
>
>
> ## Comparison with SOTA results
>
> Our result shows great improvement, especially with the impoverished 1-shot setting. Using generated samples, AlignDiff nearly doubles the novel IoU for generalized few-shot segmentation (note that this is a **different task setting than conventional few-shot segmentation**). We also demonstrate that AlignDiff is able to improve the novel IoU on the FSS-1000 dataset, greatly outperforming both real-sample-only baseline and grounded diffusion. Reviewer ofKT and Reviewer bS35 both pointed out that our method **significantly improve the existing few-shot segmentation methods**.
>
> The reviewer provided a reference to SCCAN [A] from ICCV’23, a work that deals only with the conventional few-shot segmentation setting. The main difference between conventional few-shot segmentation and the generalized few-shot segmentation is that, the generalized few-shot segmentation requires us to deal with a more challenging case where all classes need to be segmented.
>
> Nevertheless, in response to the reviewer’s requests, we run a small-scale experiment where we combine AlignDiff and SCCAN and demonstrates that SCCAN+AlignDiff outperforms plain SCCAN and achieves SOTA results on the Pascal-5i dataset under the 1-shot case that the reviewer requested. **This also demonstrates that our method is a general model-agnostic data augmentation method that can be integrated with many models to consistently improve their performance.**
>
> | Method  | Pascal-5-0 IoU | Pascal-5-1 IoU | Pascal-5-2 IoU | Pascal-5-3 IoU | Mean |
> | - | - | - | - | - | - |
> | SCCAN [A] | 69.1 | 74.0 | 66.3 | 61.6 | 67.8 |
> | SCCAN [A] +AlignDIff | **71.0** | **74.8** | **66.5** | **63.6** | **69.0** |
>
> We also added reference to SCCAN in the revised manuscript.
>
> ## Usage of terms
>
> We thank the reviewer for the comments. We use the term ‘synthetic’ data instead of ‘generated’ data following previous works [B, C]. To further clarify, we add terms ‘synthetic/generated’ in the introduction in the hope that they clearly explain that we aim at generating samples.
>
> We chose the term ‘out-of-distribution’ to highlight the misalignment of the distribution of generated samples and the distribution of synthetic samples. Out-of-distribution refers to generation of data points that fall outside the true data distribution, which we attempts to fix in this work. To avoid confusion, we revise the introduction section and give a definition of ‘OOD generation’ the first time we use this term.
>
> ## Applicability of AlignDiff
>
> Our method is not limited to a specific latent diffusion model, such as Stable Diffusion. In particular, our method makes two assumptions of the underlying diffusion model.
>
> The normalized masked textual inversion method assumes that the diffusion model is differentiable from the final synthesized image to the text encoder.
> The semi-supervised mask generation process requires generation of coarse masks. In particular, we generate coarse masks by exploiting the cross-attention layer within the diffusion model.
>
> AlignDiff is applicable to any diffusion model that satisfies the above two properties, such as this work [D] that extends latent diffusion models to support synthesis with spatial-temporal consistency.
>
> ## References
>
> [A] Xu, Qianxiong, et al. "Self-Calibrated Cross Attention Network for Few-Shot Segmentation." ICCV. 2023.
>
> [B] Li, Ziyi, et al. "Open-vocabulary object segmentation with diffusion models." ICCV. 2023.
>
> [C] Wu, Weijia, et al. "Diffumask: Synthesizing images with pixel-level annotations for semantic segmentation using diffusion models." ICCV. 2023.
>
> [D] Blattmann, Andreas, et al. "Align your latents: High-resolution video synthesis with latent diffusion models." CVPR. 2023.

---

### Official Review · Reviewer_bS35 · 2023-11-02

**Soundness:** 3 good
**Presentation:** 2 fair
**Contribution:** 3 good
**Rating:** 5
**Confidence:** 4

**Summary:**

This paper proposes to use the pretrained diffusion models to augment the training set for few-shot semantic segmentation. Specifically, this paper provides an algorithm that generates novel instances from the diffusion model conditioned on a few available training data with annotated semantic segmentation masks and class names. Experiments on standard benchmarks such as FSS-1000 demonstrate the proposed data augmentation method improves the overall performance of existing few-shot segmentation methods.

**Strengths:**

1. It is technically sound and interesting to use pretrained image generative models for data augmentation. This paper pinpoints three major types of misalignment between the synthetic distribution and the target data distribution when a naive text-conditioned image generation method is applied and proposes a simple solution per aspect.
2. Experiments demonstrate that the proposed data augmentation can significantly improve the existing few-shot segmentation methods, especially on novel categories.
3. This paper is well-written and easy to follow.

**Weaknesses:**

1. Even though the proposed data augmentation method improves the overall performance, the synthetic data can be harmful for many categories (see Fig. A.2). It is worth a more thorough analysis regarding this issue. For example, is that because the synthetic data misaligned with the target distribution, just like the text-conditioned image generation baseline?
2. As stated in the limitation section, the proposed method degrades when the gap between the target distribution and the distribution covered by the generative model is large (e.g., medical images). However, I believe the few-shot semantic segmentation would be mostly useful for rare categories that are hard to collect enough instances for training a segmentation model. For common classes (e.g., chairs, sofa, boat) of which plenty of images can be found, it is hard to justify the necessity of using synthetic data. How is the performance of the proposed method on rare objects (e.g., the rare species from the iNaturalist dataset)?

**Questions:**

See the weakness section.

---

> ### Author Response · Authors · 2023-11-21
> **Response to reviewer bS35**
>
> We thank the reviewer for the valuable comments. Below we address all the concerns one by one.
>
> ## More analysis
>
> Thanks for the insightful comments. We provide a figure in the supplementary material to analyze the failure cases of AlignDiff. We use the failed categories from Figure A2 with the FSS-1000 dataset under the few-shot setting.
>
> In summary, since AlignDiff conditions the generation process using texts and the provided real sample, when both text conditioning and the masked textual inversion fail, AlignDiff may fail to generate good examples. For qualitative figures and more detailed analysis, we kindly refer the reviewer to Fig. A13 in the supplementary material.
>
> ## Generalization to hard categories
>
> Though AlignDiff may not generalize well to domains where domain-specific knowledge is required (e.g., medical images), it is able to work on hard categories that plain text conditioning fails. In fact, this is our intention to evaluate AlignDiff on the FSS-1000 dataset, because FSS-1000 contains many rare categories.
>
> To validate this point, we provide more qualitative samples in the supplementary material, which can be found in Fig. A4 of the supplementary material. Since iNaturalist does not provide segmentation annotations, it does not work with AlignDiff. Therefore, we visualize some rare categories of fine-grained species of animals from the FSS-1000 dataset. Notice how AlignDiff is able to capture the fine-grained details of textures in these species; while text-conditioned Stable Diffusion generates irrelevant samples.

---

### Official Review · Reviewer_ofKT · 2023-11-04

**Soundness:** 3 good
**Presentation:** 3 good
**Contribution:** 2 fair
**Rating:** 6
**Confidence:** 4

**Summary:**

- The paper aims to synthesize training images and masks of novel categories to augment few-shot learning.

- They identify three issues with directly using text-conditioned stable diffusion model -- failure on OOD classes, object-centric bias of stable diffusion and coarse mask generation. These are referred to as instance, scene and annotation level misalignments.

- Failure on OOD is addressed using normalized masked textual inversion. Object-centric bias is mitigated using copy-paste augmentation, and coarse masks are refined by updating the segmenter using semi-supervised learning.

- Experiments in the GFSS (Pascal-5i, COCO-20i datasets) and FSS (FSS-100 dataset) settings show impressive results.

**Strengths:**

- The paper is well-written and well-organized
- The identification of issues arising in applying text-to-image models for data scarce few-shot semantic segmentation setting is valuable
- Even though the solutions provided to each of the issues are not novel, they are simple and well-proven methods
- Extensive ablation study is provided in the appendix (both qualitative and quantitative)

**Weaknesses:**

- Limited comparison on the FSS-1000 dataset
- Evidence of the finding that treating the learnable embedding as an adjective leads to a faster and more stable training convergence, is missing
- A clear discussion of time taken by each step (textual inversion, semi-supervised mask generation), and comparison of total time with Grounded Diffusion (which is the only alternate baseline)
- The evaluation setup describes that DiffuMask without their prompt engineering is used, but it it missing from Table1 and Table2

**Questions:**

- Do the methods compared with in Table1 and Table2 also use base classes, or is it extra information used in this approach? (for copy-paste and semi-supervised learning parts)
- An ablation on the amount of generated samples would be valuable in practice
- See weaknesses

---

> ### Author Response · Authors · 2023-11-21
> **Response to reviewer ofKT**
>
> We thank the reviewer for the valuable comments. Below we address all the concerns one by one.
>
> ## More results on the FSS-1000 dataset
>
> To illustrate AlignDiff is a model-agnostic approach. we introduce an additional baseline FSS model [A] and compare its performance on the FSS-1000 dataset with and without generated data from AlignDiff and GD. AlignDiff consistently improves the performance of [A] on the FSS-1000 dataset under 1-shot setting.
>
> | Method  | Overall IoU |
> | - | - |
> | HSNet (1-shot) | 86.5 |
> | VAT-R101 [A] (1-shot) | 90.0 |
> | HSNet + GD (1-shot) | 81.4 |
> | VAT-R101 [A] (1-shot) + GD | 84.7 |
> | HSNet (1-shot) + AlignDiff (Ours)   | 88.3 |
> | VAT-R101 (1-shot) [A] + AlignDiff (Ours) | 90.8 |
> | HSNet (5-shot) | 88.5 |
> | VAT-R101 [A] (5-shot) | 90.6 |
>
> ## Validating the advantage of learning “adjective” embedding over nouns
>
> We present qualitative and quantitative results that prove the superiority of learning an embedding for adjectives over nouns. As we have mentioned in the paper, when only a limited number of samples are available, learning an adjective embedding leads to more stable performance, as the additional nouns serve as a regularization that keeps the generation process to be related to the semantic concept.
>
> We provide qualitative samples of learning adjective and noun embedding for the same 1-shot example in Fig. A11 in the supplementary material of the revised paper. In particular, we notice that optimizing noun embedding may result in degenerate samples when only a single example with a small object is available; whereas optimizing adjective embedding works in this case.
>
> In addition, we also provide results on a small-scale experiment on Pascal-5-3.
>
> | Method  | Overall IoU |
> | - | - |
> | Baseline - learn noun | 39.9 |
> | Ours | 41.5 |
>
> ## Timing of each step
>
> Below we provide the timing of each component in our method. The number is measured using a single RTX3090 GPU and with 512 x 512 resolution.
>
> For a novel category with a few examples, AlignDiff operates in two stages. In the first stage, AlignDiff is conditioned on a few examples and uses the proposed normalized masked textual inversion method to optimize textual embeddings, which takes ~5 minutes to complete.
>
> In the second stage of sample generation, AlignDiff performs
>
> Diffusion generation: ~4s per image; (with DDIM sample; stable diffusion is slow)
> Coarse mask generation: <0.1s per image
> scoring stage: <0.1s per image
> pseudo-labeling stage: <0.05s per image.
>
> Note that the two-step scheme makes AlignDiff suitable for practical applications. The inversion overhead in the first step can be effectively amortized by the diffusion generation step in the second stage.
>
> ## Evaluation of DiffuMask without prompt engineering
>
> Thanks for the question. The main reason why we did not make a comprehensive comparison with DiffuMask without prompting is because of the expensive GPU hours it required (The inefficient noise learning process in DiffuMask requires thousands of GPU hours to generate samples on the COCO dataset).
>
> As an alternative, we present a smaller-scale study that compares our method with DiffuMask on a split of the Pascal-5i dataset. The table can be found in Table A2 in the supplementary material. We have also updated our paper to refer readers to Table A2 for better clarity. In summary, DiffuMask generates masks of similar quality with AlignDiff on common categories, but it comes at a much higher cost with the costly noise learning process.
>
> ## Clarification of base class information use
>
> Thanks for the question. To clarify, to ensure a fair comparison, some other baseline methods also use information from the base classes. For instance, GD trains its zero-shot segmenter using image-mask samples from the base dataset. Also, GAPS+GD uses samples from the base class. We observe that AlignDiff shows improvement against baselines that also use information from the base class, which validates the effectiveness of our method.
>
> ## Plot of performance w.r.t. number of generated samples
>
> Thanks for the question. We provide a plot of the novel IoU on a split of the Pascal-5i dataset in the revised PDF. The figure can be found in Fig. A12 in the revised supplementary material. Note that the performance saturates around 500-1,000 samples.
>
> References
>
> [A]: Hong, Sunghwan, et al. "Cost aggregation with 4d convolutional swin transformer for few-shot segmentation." ECCV. 2022.

---

### Author Response · Authors · 2023-11-23
**General response to all reviewers**

We thank all reviewers for their time and feedback. We greatly value the comments provided by the reviewers and have endeavored to address all their questions systematically.

Summary of our strength: Our work is 1) well-written and well-organized (reviewers ofKT, bS35, ZQEG), 2) identifies issues arising in applying text-to-image models (reviewers ofKT, bS35), and 3) with good experimental results (reviewers ofKT, bS35).

In particular, we have provided additional explanations and experimental results in the revised paper PDF on the following:
- We provided an additional figure to report the performance with respect to the number of generated samples used in Fig. A12.
- We provided a figure of uncommon animal species images in Fig. A4 to illustrate the effectiveness of our method to synthesize for uncommon classes.
- We provided a figure for analyzing the failure cases of AlignDiff in Fig. A13.
- We revised the introduction section and the related work section to clarify the task setting, terminology, and cite a related work in FSS and an alternative inversion method, Dreambooth.

---

### Meta-Review · Area_Chair_CZhD · 2023-12-08

**Metareview:**

This paper proposes a method to synthesize training images and masks for few-shot image segmentation. The author rebuttal has partially addressed points raised in the reviews, but overall the reviewer recommendations remain negative despite the additional results and clarifications. The single reviewer giving a positive recommendation did not convince the other reviewers in changing theirs. A shared concern regards the level of novelty and contribution brought by the paper.

**Justification For Why Not Higher Score:**

Only a single reviewer makes a positive recommendation (6), other reviewers have maintained or lowered their score (5,5,3). A shared concern is the level of novelty the paper brings.

**Justification For Why Not Lower Score:**

N/A

---

### Decision · Program_Chairs · 2024-01-16

Reject